

# Using an integrated hydrological model to estimate the usefulness of meteorological drought indices in a changing climate

Diane von Gunten[1], Thomas Wöhling[1,2,3], Claus P. Haslauer[1], Daniel Merchán[4], Jesus Causapé[4], and Olaf A. Cirpka[1]

[1]University of Tübingen, Center for Applied Geoscience, Hölderlinstr. 12, 72076 Tübingen, Germany
[2]Technische Universität Dresden, Department of Hydrology, Bergstr. 66, 01069 Dresden, Germany
[3]Lincoln Agritech Ltd., Ruakura Research Centre, Hamilton, New Zealand
[4]Geological Survey of Spain – IGME, C/ Manuel Lasala no. 44, 9B, Zaragoza, 50006, Spain

*Correspondence to:* Olaf A. Cirpka (olaf.cirpka@uni-tuebingen.de)

**Abstract.** Droughts are serious natural hazards, especially in semi-arid regions. They are also difficult to characterize. Various summary metrics representing the dryness level, denoted drought indices, have been developed to quantify droughts. They typically lump meteorological variables and can thus directly be computed from the outputs of regional climate models in climate-change assessments. While it is generally accepted that drought risks in semi-arid climates will increase in the future, quantifying this increase using climate model outputs is a complex process which depends on the choice and the accuracy of the drought indices, among other factors. In this study, we compare seven meteorological drought indices that are commonly used to predict future droughts. Our goal is to assess the reliability of these indices to predict hydrological impacts of droughts under changing climatic conditions. We simulate the hydrological responses of a small catchment in northern Spain to droughts in present and future climate, using an integrated hydrological model, calibrated for different irrigation scenarios. We compute the correlation of meteorological drought indices with the simulated hydrological times series (discharge, groundwater levels, and water deficit), and we compare changes in the relationships between hydrological variables and drought indices. While correlation coefficients are similar for all tested land-uses and climates, the relationship between drought indices and hydrological variables often differs between present and future climate. Drought indices based solely on precipitation often underestimate the hydrological impacts of future droughts, while drought indices that additionally include potential evapotranspiration sometimes overestimate the drought effects. In this study, the drought indices with the smallest bias were: the rainfall anomaly index, the reconnaissance drought index, and the standardized precipitation evapotranspiration index. However, the efficiency of these drought indices depends on the hydrological variable of interest and the irrigation scenario. We conclude that meteorological drought indices are able to identify the timing of hydrological impacts of droughts in present and future climate. However, these indices are not capable of estimating the severity of hydrological impacts of droughts in future climate. A well-calibrated hydrological model is necessary in this respect.





## 1 Introduction

In semi-arid regions, droughts are a serious natural hazard, often causing tens of millions of euros of damage (Gil et al., 2011). In northern Spain, for example, drought severity has increased in the last decades (Hisdal et al., 2001) and is expected to increase further in the next 50 years (Bovolo et al., 2010), as a result of the ongoing increase in global mean temperature (e.g., Meehl et al., 2007). More severe droughts will negatively impact the region, notably the agricultural sector (Stahl et al., 2015).

Droughts have a wide range of impacts, and are often difficult to define. They have been classified in four main categories (Mishra and Singh, 2010; Samaniego et al., 2013; Wilhite and Glantz, 1985):

- *meteorological* droughts defined by a lack of precipitation over a certain period of time for a certain region,

- *hydrological* droughts defined by a reduced surface and subsurface water availability for a given water resource,

- *agricultural* droughts defined by a period of declining soil moisture and reduced crop yields,

- and *socio-economical* droughts defined by a failure of water-resources management to meet the supply and demand of water (taken as an economic good).

In order to quantitatively describe drought levels, about 150 different drought indices have been developed (Zargar et al., 2011). A drought index is a scalar composed of one or more measured variables affected by dry and wet periods. In the case of meteorological drought (which is the focus of this study), typical variables considered for the calculation of drought indices are precipitation and potential evapotranspiration.

In addition to the identification of drought periods, these meteorological drought indices are also good indicators for various droughts impacts in present climate, based on the results of a range of studies. For example, text-recollection of droughts, such as newspaper articles, are linked with different drought indices, indicating a relationship between the social impacts of droughts and drought-index values (Bachmair et al., 2015). Crop yields are also correlated with drought indices in different climatic regions (e.g., Quiring and Papakryiakou, 2003; Mavromatis, 2007). Moreover, Vicente-Serrano et al. (2012a) analyzed the correlation between six drought indices and environmental variables, such as stream flow, tree rings widths, and soil moisture. A significant correlation between the studied environmental variables and the drought indices was found. The correlation between groundwater levels and drought indices seems to be smaller than for other drought impacts, but it was still noticeable (Kumar et al., 2015).

Hence, meteorological drought indices are correlated with hydrological and agricultural impacts of meteorological droughts. Consequently, they are also correlated with hydrological or agricultural droughts. Many of the drought impacts cited above, such as changes in groundwater levels or discharge, could also be conceptualized as an indicator of hydrological or agricultural droughts. For example, groundwater levels could be transformed to a drought indicator such as the standardized groundwater level index (SGI, Bloomfield and Marchant (2013)) to identify hydrological droughts (Kumar et al., 2015). Indeed, hydrological impacts of drought and hydrological drought indices are often two perspectives of the same drought event. The viewpoint of this study is that changes in environmental variables are introduced by non-stationary meteorological forcings, i.e., that hydrological



changes are a consequence of meteorological droughts. Therefore, we will not use hydrological variables to define droughts. We explain our motivations for this choice in Sect. 2.1.

The relationship between meteorological drought indices and drought impacts is valid for many drought indices in present climate, including simpler indices using one input variable, such as precipitation. However, the suitability of drought indices

has not been tested under a changing climate. The ongoing increase in air temperature was not taken into account. Because climate change will probably impact drought intensity and frequency (e.g., Dai, 2011), various studies have aimed at predicting future changes in dry periods using drought indices based on the output of regional or global climate models. An assumption of these studies is that drought indices perform similarly in present and future climate. Our aim is to test this hypothesis. That is, we will test the capability of meteorological drought indices to predict hydrological impacts of drought under a changing

climate.

A large number of drought indices have been used in recent climate-impact studies. For instance, the standardized precipitation index was often used to study future droughts (e.g., Leng et al., 2015; Masud et al., 2015; Tue et al., 2015; Zarch et al., 2015). However, several studies used other indices, as the reconnaissance drought index (e.g., Kirono et al., 2011; Zarch et al., 2015), the standardized precipitation evapotranspiration index (e.g., Kim et al., 2014; Masud et al., 2015), the effective drought

index (e.g, Park et al., 2015), or the Palmer drought severity index (e.g., Burke et al., 2006), among others. The choice of the drought index can have an important impact on the results. For example, Kim et al. (2014) and Park et al. (2015) predicted future droughts over Korea in the next century using very similar climate scenarios. While Kim et al. (2014) projected an increase in the severity of droughts in this region, Park et al. (2015) projected a more complex spatial pattern and a possible decrease in drought severity in coastal regions. A possible reason for these contradictory results is that Park et al. (2015) used a drought

index based on precipitation only, while Kim et al. (2014) used an index which considers both potential evapotranspiration and precipitation. Precipitation-based drought indices, such as the effective drought index (EDI) or the standardized precipitation index (SPI), tend to work well in present climate. However, they may be inadequate to predict climate-change effects because they neglect the increase in potential evapotranspiration, resulting in a possible under-estimation of the intensity of future droughts (Dubrovsky et al., 2009; Vicente-Serrano et al., 2009, 2015; Zarch et al., 2015).

To study the validity of drought indices in future climate, we chose seven well-known drought indices, which can be computed from the output of climate models, such as precipitation, temperature or potential evapotranspiration. We investigate the ability of these indices to predict hydrological variables under drought conditions: groundwater heads, discharge at the catchment outlet, and water deficit of the crops, under present and (projected) future climate conditions. These three metrics address different hydrological effects of droughts of high ecologic and/or economic relevance. Reduced stream discharge can

deteriorate the ecological status of the stream because the stream temperature and the concentrations of contaminants increase with decreasing discharge. In the most extreme case, the stream falls dry. The drawdown of groundwater heads is of high economic relevance when groundwater is pumped for water supply and irrigation which, however, is not the case in the studied catchment. Groundwater levels also control low flows in gaining streams. Finally, the water deficit of the crops, that is, the difference between transpiration under conditions when enough water is available and the actual transpiration, is a simple metric

of water stress experienced by the crops, which may diminish crop yields.



A fully-integrated hydrological model of a small catchment, the Lerma catchment, in north-east Spain, is used to simulate the hydrological responses to the meteorological forcing. This catchment has recently undergone a monitored transition from rainfed to irrigated agriculture, in which the irrigation water is imported from the Yesa reservoir located outside of the catchment (Merchán et al., 2013). The model was calibrated under different irrigation conditions (von Gunten et al., 2014), which

increases our confidence in its ability to predict the hydrological responses to changes in (meteorological and land-use) forcing. We use these different land-use/irrigation schemes to test the different drought indices. The outputs from a weather generator, representing present and future climate, are used as meteorological inputs to the model and for the computation of the drought indices.

The remainder of this paper is structured as follows: First, we present the methodology used in this study. Specifically, we

briefly describe the study area, the hydrological model, the climate scenarios, the irrigation scenarios, and the drought indices. Secondly, we compare the frequency distribution of drought indices computed from measurements and based on the outputs of the weather generator. Next, we summarize an analysis of the correlation coefficients between hydrological variables and drought indices for two different land-uses (with/without irrigation), and for present and future climate scenarios. Afterwards, we investigate changes in the relationship between these drought indices and the hydrological variables. We then use these

results to predict relevant changes in drought risks in the study area in future climate. Finally, we discuss the usefulness of drought indices in climate-impact studies.

## 2 Methods

### 2.1 Overview

The main objective of this paper is to test the suitability of several meteorological drought indices to estimate the impacts

of climate change on the water cycle of a small catchment. Seven drought indices, described in Sect. 2.6, are investigated. The information on drought severity (as computed by these indices) is compared to three simulated hydrological impacts of drought: (1) the mean annual discharge at the outlet, (2) the mean annual hydraulic heads in 12 observations wells of the local aquifer, and (3) the water deficit ($WD$), which is a simplified representation of how well the water demand of the crops can be met (Abrahao et al., 2011):

$$WD\,[\%] = 100 \times \frac{ET_c - AET}{ET_c} \tag{1}$$

where $ET_c$ is the annual crop evapotranspiration under standard conditions (Allen et al., 2000), and AET is the simulated actual evapotranspiration, calculated on the yearly time scale.

The time series of the drought impacts listed above are obtained using the outputs from a calibrated, integrated, pde-based, hydrological model (Sect. 2.3) forced by present and future meteorological time series (Sect. 2.4), and daily irrigation scenarios

(Sect. 2.5). Five climate scenarios (one based on present climate and four based on the projections of regional climate models) and three irrigation scenarios are constructed and combined with each other in our simulations. The length of the simulation





is 180 years for each combination of (present and future) climate and irrigation scenarios. This is equivalent to a total 2700 simulated years. From these 2700 simulated years, we extract time series of discharge, hydraulic heads, and water deficit.

In this study, the time series of these three hydrological variables are directly used to represent the drought impacts on hydrology. We do not consider indicators of hydrological droughts such as the standardized groundwater level index (SGI,

Bloomfield and Marchant (2013)) or the standardized streamflow index (SSI, Vicente-Serrano et al. (2012b)). SGI and SSI are hydrological drought indices representing drought events using the normalized changes in hydraulic heads and discharge, respectively. We decided not to use these indices here because the focus of this study is on meteorological droughts. In this context, changes in hydrological variables during dry periods are the consequence of a drought and not an indicator of a drought situation. Moreover, we restrict our investigation to drought indices which can be computed from the outputs of regional or

global climate models (precipitation, temperature, etc.), which would not be possible for SGI and SSI. Finally, no drought indices similar to SSI or SGI exist for water deficit. Therefore, a transformation from time series inputs to drought indices would not have been easily possible for this variable. For these reasons, only time series of the hydrological variables are investigated in this study.

The three hydrological time series are compared to the time series of meteorological drought indices (Sect. 2.8): We first

compute the Peason correlation coefficient between the drought indices and the hydrological variables. Next, we analyze changes in the (assumed) linear relationship between hydrological variables and drought indices. These comparisons are repeated in present and future climate for the different irrigation scenarios. A suitable drought index for climate-change studies would have a large correlation coefficient with all hydrological variables and the relationships between this index and the hydrological variables would be identical in present and future climate. The results of these quantitative studies are presented in

Sect. 3, while we discuss additional aspects of using meteorological drought indices in climate-impact studies in Sect. 4.

## 2.2  Study area

The Lerma catchment is situated within the Ebro basin in Spain with an altitude varying between 330 and 490 masl., and an area of ~7.3 km$^2$ (Fig. 1). Its climate is classified as semi-arid, with a mean precipitation of ~400 mm/year (2004-2011) and a mean potential evapotranspiration rate of ~1300 mm/year (2004-2011) (Merchán et al., 2013). Precipitation and temperature have

been measured since 1988 at the meteorological station of Ejea de los Caballeros (~5 km north of the study area). Radiation, wind, and relative humidity have been measured since 2003. Annual precipitation is highly variable, ranging from 268 mm/year to 558 mm/year (2004-2011). Because of the limited water resources, drought is a serious natural hazard in the region (Bovolo et al., 2010).

The catchment underwent a rapid transition from non-irrigated to irrigated agriculture between 2006 and 2008. The majority

of the fields within the catchment are now irrigated, with an irrigation water volume of $2.1 \cdot 10^6$ m$^3$ in 2011 (Merchán et al., 2013). This transition was closely monitored and monthly hydraulic head data, daily discharge, crop types, and daily irrigation volume are available. In addition, a vertical-electrical-sounding campaign (Plata-Torres, 2012) was conducted to better understand the local geology. Two main hydrologically relevant layers were identified: The top layer is composed of clastic and unconsolidated Quaternary deposits and forms a shallow aquifer. Underneath lies an aquitard composed of lutite and marl-





stones (Fig. 2). Soils are relatively shallow, with depths below ground surface ranging between 0.3 and 0.9 m (Beltrán, 1986), and are classified as inceptisols.

## 2.3 Hydrological model

To simulate the hydrological response of the Lerma catchment, we use HydroGeoSphere (Therrien, 2006), a three-dimensional,
fully-coupled, integrated hydrological model, based on partial differential equations. In HydroGeoSphere (Therrien et al., 2010), water flow in the variably-saturated sub-surface is modelled using the three-dimensional Richards' equation, while overland flow is simulated by the diffusive-wave approximation of the Saint-Venant equations. We use the Mualem-van Genuchten parametrization (van Genuchten, 1980) to relate relative permeability and water saturation to capillary pressure in the vadose zone. The surface and subsurface domains are coupled using a dual-node approach, where the coupling between the domains is
conceptualized as a virtual thin layer of porous material. The model choice is based on the necessity to model the transition to irrigation, which has a large impact on the hydrology of the catchment. Moreover, HydroGeoSphere allows to simultaneously study the impact of droughts on the surface and subsurface components of water flow. The underlying equations have been reviewed by von Gunten et al. (2014, 2015) and are not repeated here.

The conceptual model of our study area and its calibration have also been presented by von Gunten et al. (2014) and thus
are only presented here briefly. We divide the sub-surface catchment in six zones, two zones representing the aquitard, one representing the aquifer, and three representing the different soil zones (Fig. 2). The model parameters are homogeneous in each zone and the saturated hydraulic conductivity is one order of magnitude smaller in the vertical direction than in the horizontal one. The surface domain is divided into 55 zones, representing the different farm fields. Daily irrigation volume, Manning's parameters, seasonal leaf area index, and rooting depth are specified separately for each surface zone, based on
crop types and irrigation data. Precipitation is given as daily input, apart from days with intense rainfall (>25mm/day). In this case, precipitation data is given as a 3-hour mean during summer and spring, and as a 9-hour mean during autumn and winter, to mimic intense convection events (von Gunten et al., 2014), which are frequent in the region. A no-flow boundary condition is assumed at the lateral and the bottom boundaries of the sub-surface domain. Critical flow depth is used for the lateral boundaries of the surface flow domain.

We calibrated the parameters of the model using three computational grids of increasing resolution (von Gunten et al., 2014). The calibrated parameters are the hydraulic conductivity in all zones, apart from the "weathered aquitard" zone (Fig. 2), the porosity of the aquifer, and the van-Genuchten parameters of the soil zones. The calibration period is from 2006 to 2009 and the validation period is from 2010 to 2011. The model is calibrated on the measured discharge at the outlet and on the hydraulic heads in eight observation wells (twelve observation wells were used during validation). The model reproduces the
measurements satisfactorily (von Gunten et al., 2014). For example, the Nash-Sutcliffe efficiency (Nash and Sutcliffe, 1970) of discharge is of 0.74 during the calibration period and of 0.92 during the validation period. The model performs similarly well under all irrigation conditions. Because the model was able to reproduce the response in both discharge and groundwater tables to the changes in irrigation practice, we are confident that it can also predict the response to changes in meteorological forcing projected by climate models.



## 2.4 Climate scenarios

The climate scenarios used in this study have been presented by von Gunten et al. (2015) and are thus only summarized here.

Our future climate scenarios cover the time period of 2040-2050, using the A1B IPCC emission scenario (Nakićenović et al., 2000). They are based on 4 regional climate models from the ENSEMBLES project (van der Linden and Mitchell, 2009)
driven by two global climate models (Table 2). As it is not advisable to use the direct outputs from climate models as input for a small-scale hydrological model (Prudhomme et al., 2002), we have downscaled the outputs from the climate models using a weather generator, i.e., a statistical model reproducing the characteristics of the observed climatic time series (Srikanthan and McMahon, 2001). We calibrated the weather generator using the observed time series of the closest meteorological station (Ejea de los Caballeros). Then, the parameters of the weather generator were modified using the differences between the
control and future simulations of the regional climate models. These change factors, described in Burton et al. (2010), are an indication of future changes of the mean and variability of precipitation, temperature, radiation, and relative humidity. The weather generator is run using the updated parameters to create the future climate scenarios. In this study, we use the RainSim weather generator for precipitation (Burton et al., 2008) and the EARWIG weather generator for potential evapotranspiration (Kilsby et al., 2007).

The downscaling of climate model outputs is a complex task and the choice of a particular downscaling method can have a large impact on the results (Holman et al., 2009). Our study is not an exception and the downscaling process presented here might introduce uncertainties in the climate scenarios. We have mitigated this issue using three different approaches: a) We prepared both present and future time series of meteorological inputs using the weather generator. Hence, the potential bias resulting from the weather generator is reproduced in the present and future time series. b) The time series of present
precipitation and potential evapotranspiration have been extensively tested against measurements to control the quality of the weather generator outputs (von Gunten et al., 2015). c) We compared the future time series of precipitation and potential evapotranspiration downscaled with the weather generator with the corresponding time series downscaled with a simpler bias correction method (Li et al., 2009). The time series were found to be generally similar regardless of the downscaling method (von Gunten et al., 2015). Hence, we consider that the quality of the climate downscaling in our study is acceptable.

The chosen downscaling procedure has the advantage of producing longer time series, compared to the relatively short (23 years) climate record in the Lerma catchment. Moreover, it reproduces future changes in the precipitation variability, and not only in the precipitation mean, which is an important criterion when studying future droughts.

Future precipitation (Fig. 3) is predicted to decrease in summer and spring (between 3% and 39% of the current precipitation, depending on the regional climate model). In winter and autumn, an increase in precipitation is predicted (between 1% and
55%). Change in total annual precipitation depends on the regional climate model. MPI and UCLM predict a wetter future, while ETHZ and METO predict a dryer one (see Table 2 for the references of the regional climate models). The coefficients of variation increase in spring (between +3% and +6%), decrease in winter and autumn (between -0.1% and -10%), and do not show a clear trend in summer (between +5% and -5%).





Because of the higher temperature, potential evapotranspiration increases (between 9% and 22% in the annual average) in all regional climate models for all months. This increase might impact droughts, regardless of the precipitation changes.

## 2.5 Irrigation scenarios

Consistent with our earlier study (von Gunten et al., 2015), we use three irrigation (or land-use) scenarios that can be summarized as follows:

- scenario NOIRR: without irrigation and without agriculture.

- scenario PIRR: with present cropping patterns and present irrigation.

- scenario FUTIRR: with present cropping pattern, but with an updated irrigation volume to account for future climatic conditions. To create this scenario, we assume that the irrigation efficiency will not change in future climate. In addition, we assume that the increase in irrigation will only depend on the increase in potential evapotranspiration and changes in precipitation amount (see Toews and Allen, 2009).

## 2.6 Drought indices

More than 150 drought indices have been developed in the past (Zargar et al., 2011) and it would be unrealistic to include all of them in this study. Therefore, we have selected seven well-known and commonly-used drought indices, based on the reviews by Agwata (2014), Hayes and Lowrey (2007), Heim (2002), Niemeyer (2008), and Zargar et al. (2011). Our choice was guided by the required data input and the popularity of the indices in recent studies related to climate change. We present the selected indices briefly below and provide a summary in Table 1.

In this study, we generally consider meteorological drought indices that aggregate data annually. The only exceptions are the Palmer drought indices (PDSI and PHDI) whose time length depends on an empirical estimation of the start and the end of drought periods (Szép et al., 2005). We chose an annual time scale because it is often used when predicting future droughts (e.g., Kirono et al., 2011; Park et al., 2015) and because it is the most dominant precipitation cycle worldwide (Park et al., 2015).

Our definition of a drought is identical for present and future climate. Practically, we standardize the drought indices in the present climate and keep the same standardization (explained below) in the future climate. From a conceptual point of view, this is unexpected as meteorological droughts can be defined as a period of exceptionally dry conditions. If the average precipitation changes, the definition of a meteorological drought should also be changed. However, from a practical point of view, drought severity depends on the water needs and on the vulnerabilities of the social and agricultural structures. Hence, the definition of future droughts is linked to current conditions. From this perspective, using the same standardization in present and future climate is logical. Moreover, this procedure has been applied in the majority of studies on future droughts (e.g., Zarch et al., 2015).



### 2.6.1 Standardized precipitation index (SPI)

SPI (McKee et al., 1993; Svoboda et al., 2012) is a widely-used drought index (Zargar et al., 2011). To compute this index, precipitation data is first fitted to a probability distribution. We use a gamma distribution with the shape parameter $\alpha$ and the scale parameter $\beta$ (Wu et al., 2005). The fitted parameters $\hat{\alpha}$ and $\hat{\beta}$ are then used to find the cumulative probability $G(P)$ of the precipitation amount $P$ (Edwards, 1997):

$$G(P) = \int_0^P g(x)dx = \frac{1}{\hat{\beta}^{\hat{\alpha}}\Gamma(\hat{\alpha})} \int_0^P x^{\hat{\alpha}-1} e^{\frac{-x}{\hat{\beta}}} dx \tag{2}$$

where $\Gamma$ is the gamma function or $\Gamma(\alpha) = \int_0^\infty x^{\alpha-1} e^{-x} dx$. The probability of null precipitation $q$ is estimated by dividing the number of dry months by the length of the monthly time series. It is accounted for by:

$$H(P) = q + (1-q)G(P) \tag{3}$$

To compute the value of SPI, an equiprobability transformation is made from the cumulative probability $H(P)$, i.e., $H(P)$ is transferred to a standard normal random variable with a mean of zero and a variance of unity:

$$SPI = \Phi^{-1}(H(P)) \tag{4}$$

where $\Phi$ is the standard normal cumulative distribution function. SPI takes monthly precipitation as input and can be computed at various time scales, from 1 month to 24 months. In this study, we use a 12-months time scale. An SPI-value smaller than -1 indicates a dry period, and an SPI-value larger than +1 a wet period (Svoboda et al., 2012).

### 2.6.2 Standardized precipitation evapotranspiration index (SPEI)

SPEI (Vicente-Serrano et al., 2009) has been developed to account for the impact of potential evapotranspiration on droughts, especially in a changing climate. Its computation is similar to SPI. For SPEI, the difference between precipitation and potential evapotranspiration, rather than only precipitation, is used in the index computation. This time series is fitted to a probability distribution as described for SPI. A log-logistic distribution (e.g., Ashkar and Mahdi, 2006) is used here, following Vicente-Serrano et al. (2009). The sensitivity of SPEI to potential evapotranspiration is higher than other drought indices (Vicente-Serrano et al., 2015), such as PDSI or RDI (defined in Sect. 2.6.5 and 2.6.6).

### 2.6.3 Rainfall anomaly index (RAI)

RAI can be used to analyze dry or wet periods. When used to study droughts, RAI (e.g., Keyantash and Dracup, 2002) represents a ranking of yearly precipitation, compared to the most negative precipitation anomalies recorded. It is defined as follows:

$$RAI = -3\frac{P - \bar{P}}{\bar{E} - \bar{P}} \tag{5}$$





where $P$ is the annual precipitation, $\bar{P}$ the mean annual precipitation and $\bar{E}$ is the precipitation average of the ten driest years. Negative values of RAI indicate dry periods.

### 2.6.4 Effective drought index (EDI)

In contrast to the other drought indices, EDI (Byun and Wilhite, 1999) is computed using daily precipitation to better take into account the effect of precipitation variability on droughts. The effective precipitation $EP$ is calculated first:

$$EP = \sum_{n=1}^{i} \frac{\sum_{d=1}^{n} P_d}{n} \tag{6}$$

where $i$ is the summation period and $P_d$ is the precipitation of $d$ days before the end of the period $i$. We choose $i = 365$ days in our application, i.e., annual averages. EP is then normalized to calculate the EDI:

$$EDI = \frac{EP - \overline{EP}}{\sigma_{EP}} \tag{7}$$

where $\overline{EP}$ is the mean of the effective precipitation ($EP$) and $\sigma_{EP}$ its standard deviation.

### 2.6.5 Palmer drought severity index (PDSI) and Palmer hydrological drought index (PHDI)

PDSI was developed by Palmer (1965) to better consider the role of evapotranspiration on droughts and to "measure the cumulative departure of moisture supply" during dry periods. This index is composed of a simplified water balance of a basic two-layer soil model which is then compared to a reference water balance time series. It is a dimensionless number, usually ranging between -4 and +4, with negative values indicating dry periods (Keyantash and Dracup, 2002). It is widely used, especially in the United States, but it is relatively involved to calculate (Jacobi et al., 2013). In addition, it assumes a homogeneous soil type and the time window considered by the index varies depending on the weather.

PHDI (Palmer, 1965) is a variation of the previous index which has been developed to better represent hydrological droughts. To achieve this, PHDI applies the same simplified soil model as PDSI, but stricter criteria are used to define the limits of the wet and dry periods. This results in an index which reacts more gradually than the original Palmer index (Keyantash and Dracup, 2002).

In this study, we use the Matlab tool developed by Jacobi et al. (2013) to calculate PDSI and PHDI.

### 2.6.6 Reconnaissance drought index (RDI)

The RDI (Tsakiris and Vangelis, 2005) is based on the FAO aridity index $\alpha_i$, defined as:

$$\alpha_i = \frac{\sum_{j=1}^{12} P_{ji}}{\sum_{j=1}^{12} PET_{ji}} \tag{8}$$





where $P_{ji}$ is the monthly precipitation of the year $i$ and $PET_{ji}$ is the monthly potential evapotranspiration. The standardized RDI is computed as followed (Tsakiris and Vangelis, 2005):

$$RDI = \frac{ln(\alpha_i) - \overline{ln(\alpha_i)}}{\sigma_{ln(\alpha_i)}} \qquad (9)$$

where $\sigma_{ln(\alpha_i)}$ is the standard deviation of the natural logarithm of the aridity index and $\overline{ln(\alpha_i)}$ its mean.

## 2.7 Computation of potential evapotranspiration

SPEI, PDSI, PHDI, and RDI are calculated using the same expression for potential evapotranspiration ($ET_0$), sometimes referred to as reference evapotranspiration. We use the well-known FAO Penman-Monteith equation (Allen et al., 2000) in all our calculations. Some indices, for example PDSI, are often computed using simpler expressions for potential evapotranspiration that are based only on temperature, such as the Thornthwaite equation (Jacobi et al., 2013). However, we compute all indices with identical $ET_0$ to avoid an undue influence on the performance of the drought indices by the choice of $ET_0$.

The hydrological model, described in Sect. 2.3, also uses daily inputs of reference evapotranspiration as estimated by the FAO Penman-Monteith equation (Allen et al., 2000). $ET_0$ is then multiplied by a time-varying crop coefficient to account for the different crop types and their spatial distribution in the catchment. Hence, the final model input is the spatially-explicit daily crop evapotranspiration under standard conditions ($ET_c$), corresponding to the maximum evapotranspiration of each crop without water limitation. The crop coefficients are taken from Allen et al. (2000). Although $ET_c$ is used to simulate hydrological impacts, it is not used in the computation of drought indices. Here, we use $ET_0$ in all calculations. This is consistent with the approaches used in other studies. We want to mimic the typical utilization of drought indices, which are usually computed directly from meteorological data (e.g., Zarch et al., 2015). To test the impact of our assumption, we repeated the analysis presented in this paper using $ET_c$ instead of $ET_0$ (results not shown) and found very similar correlations and relationships between drought indices and hydrological variables.

The potential evapotranspiration used by the hydrological model and in the computation of drought indices is calculated from the outputs of a weather generator (Sect. 2.4). To validate the outputs of the weather generator (Sect. 3.1), time series of potential evapotranspiration are prepared, based on measured time series. More precisely, we use 23 years of precipitation and temperature (1988-2011) measured at the meteorological station of Ejea de los Caballeros (Sect. 2.2). Time series of radiation, wind, and relative humidity are also needed to calculate $ET_0$. However, these variables are only measured for the last 9 years. For the 14 years with missing data, $ET_0$ is calculated using the daily mean radiation, wind, and relative humidity averaged over the last 9 years and on the actual measurement of temperature. Differences between the usual calculation of $ET_0$ and the calculation based on averaged radiation, relative humidity, and wind are small. The Nash-Sutcliffe efficiency (Nash and Sutcliffe, 1970) between the $ET_0$ using the full data set and the $ET_0$ based on averaged data is above 0.85 for the 9 last years.



### 2.8 Methods of comparing the drought indices to predict hydrological variables

To compare how well the drought indices can predict the chosen hydrological variables in present and future climate, we use two approaches. First, we compute the Pearson's linear correlation coefficient $r$ between the time series of meteorological drought indices and the hydrological variables. Secondly, we compute changes in the coefficients of the (assumed) linear
regressions between the time series in present and future climate and the drought indices.

#### 2.8.1   Pearson's correlation coefficient

The Peason's linear correlation coefficient quantifies how well the variability in one time series can be explained by the variability of another time series, assuming a linear relationship between the two variables. In the context of this study, it indicates if the drought indices have the capability of finding periods with a discharge or hydraulic heads lower than usual and
periods with water deficit higher than usual. It is defined as follows:

$$r = \frac{cov(DI, x)}{\sigma_{DI}\,\sigma x} \qquad (10)$$

in which $cov$ is the covariance, $DI$ is the value of the drought index and $x$ is the hydrological variable under consideration. The range of $r$ is -1 to +1, where +1 indicates a perfect positive correlation, -1 is a perfect negative correlation, and a value of zero signifies no correlation.

#### 2.8.2   Linear regression

The Pearson's correlation coefficient indicates the degree of linear dependence between two variables. However, if this correlation coefficient is calculated under different climatic conditions, it does not indicate possible changes in the coefficients of the (assumed) linear dependencies. To investigate the changes in the linear dependency between the two climates, we perform a linear regression between a drought index and a hydrological variable in the present climate. Then, we use this linear
relationship to predict the hydrological variables from the same drought indices in future climate. We conduct this analysis for each combination of drought index and hydrological impact in all irrigation scenarios. By this, we aim to investigate if drought indices in future climate represent on average a similar drought (i.e., a drought with similar hydrological impacts) than in present climate. This is important because many drought studies (e.g., Kirono et al., 2011) only report changes in drought indices, implicitly assuming identical drought impacts for identical drought-index values in present and future climate. How-
ever, a drought described by a SPI-value of -1, for example, may have different consequences on discharge and water deficit in projected future climate than under current climate conditions (see Sect. 3.3).

To quantify the changes in the linear dependencies between hydrological variables and drought indices, two performance metrics were selected: The model bias $B$ and the normalized root mean square error ($NRMSE$). The model bias is the sum of the differences between the predicted and the actual values of the hydrological variable:

$$B = \sum_{i=1}^{n} V_{stat,i} - V_{mod,i} \qquad (11)$$





where $V_{stat,i}$ indicates the predicted value of discharge or water deficit based on the linear regression, $V_{mod,i}$ represents the value of the same variable predicted by the hydrological model and $n$ is the length of the time series.

The $NRMSE$ is the root mean square error divided by the standard deviation of the least-square regression in present climate $\sigma_{pres}$:

$$NRMSE = \frac{1}{\sigma_{pres}} \sqrt{\frac{\sum_{i=1}^{n}(V_{stat,i} - V_{mod,i})^2}{n}} \tag{12}$$

In present climate, the variability of the differences between the outputs from the hydrological model and the linear regression is smaller than 12% of the average difference between model outputs and the linear regression. Hence, the error of the linear model in the present climate can be considered homoscedastic, i.e, $\sigma_{pres}$ is considered constant in the subsequent analysis.

## 3 Results

### 3.1 Validation of the weather generator outputs

Because the outputs from the weather generator are used to compute the drought indices and to force the hydrological model, the weather generator must reproduce the observed characteristics of the meteorological variables. The calibration and validation of the weather generator for the statistics of precipitation and potential evapotranspiration has been presented by von Gunten et al. (2015). We extend this work by comparing the frequency distribution of the studied drought indices in the observed climate record with the corresponding frequency distribution computed from the weather generator outputs in the current climate.

All seven drought indices used in our study are normalized (Sect. 2.6) so that they can be used in different regions. If the normalization would have been carried out separately in the observed and simulated data, the frequency distributions of the drought indices would be similar, regardless of the similarity of the time series. To provide a meaningful comparison, we compute the normalization on the simulated data (weather generator) and we use the same normalization for the observed data (current climate record).

To compute each drought index, we use the measured time series, which has a length of 23 years (1988-2011). In addition, we compute the drought indices using the simulated data. To get a comparable length between measured and modeled data, the time series of drought indices based on the weather generator are separated into 15 periods with a duration of 23 years each (totaling 354 years). The final length of this time series is chosen such that it is about twice the length of the hydrological simulations (180 years). We then prepare 15 empirical cumulative distribution functions (*ecdf*) based on the outputs of the weather generator and compare them with the *ecdf* based on the current observed climate record (Fig. 4).

The *ecdf* of all drought indices based on measurements fall into the region defined by the 15 modeled *ecdf*. Hence, differences between the observed and simulated data were small, compared to the difference between the 15 modeled *ecdf*. In addition, we used a 2-sided Kolmogorov-Smirnov test to compare the time series based on modeled and measured data. This test (e.g., Hazewinkel, 2001) is a non-parametric statistical test which quantifies the maximum distance in cumulative probability





between two distributions and tests how likely it is that the two samples are drawn from the same distribution. All drought indices pass this test, i.e., the null hypothesis of identical *ecdf* between measured and simulated data is not rejected at a 5% significance level. Therefore, the drought indices based on the time series of the weather generator outputs are showing a reasonable agreement with the observed time series to be used in present climate. Weather generators are commonly operated

to produce time series of future hydro-meteorological variables (e.g., Burton et al., 2010) and we are also confident to use the weather generator to produce future time series of drought indices.

## 3.2 Correlation coefficients between drought indices and hydrological variables

In this section, we analyze the correlation between the different drought indices for the 180 years of each scenario and the corresponding simulated mean annual discharge, water deficit, and hydraulic heads. For this purpose, we use the Pearson's

linear correlation coefficient $r$ between the drought indices and the hydrological variables (Sect 2.8.1). We conduct the same analysis for present and future climate, and for the different irrigation scenarios. Here, we present only the main results of this comparison (details are available in the supplementary material).

In summary, the correlation coefficients between the hydrological variables and the drought indices are similar for all irrigation scenarios in present and future climate. For example, let us consider the correlation coefficients between drought

indices and discharge (Fig. 5). In present climate, SPEI, RDI, and RAI have the highest correlation with discharge in the PIRR scenario ($0.77 < r < 0.80$) as well as in the NOIRR scenario ($0.81 < r < 0.83$). These indices also have similar correlation coefficients in future climate ($0.79 < r < 0.84$). If we consider the correlation of a particular drought index with discharge over all climate/irrigation scenarios, the differences in $r$ is $< 0.1$.

Water deficit exhibits a similar behavior as discharge when correlation coefficients are examined. When the absolute values

of correlation coefficients are large in present climate, they will be similarly large in future climate or in another irrigation scenario. SPEI, RDI, and RAI have the largest correlation coefficients with water deficit in all scenarios ($0.78 < |r| < 0.81$).

Correlation coefficients between drought indices and groundwater heads in a particular observation well are similar for all drought indices considered. However, the correlation coefficients are very different from one observation well to another (see supplementary material for more information).

## 3.3 Linear regressions between hydrological variables and drought indices

The previous section has shown that the linear correlation between drought indices and hydrological variables is relatively similar under all climatic and irrigation conditions. Hence, a particular drought index is able to identify the dry periods in present and future climate. However, this does not indicate whether the droughts in future climate have similar hydrological impacts than those in present climate. Correlation coefficients quantify how well a relationship between two variables can be

expressed by an (assumed) linear equation, without considering the actual coefficients of the linear equation. The latter are commonly evaluated by linear regression.

Identifying changes in the regression coefficients of the relationships between drought-indices and hydrological variables is important when making hydrological predictions based on meteorological drought indices in a changing climate. Only when





the regression coefficients do not change, the same value of a drought index has the same hydrological impact. Towards this end, we compare changes in the (assumed) linear regressions between drought indices, and discharge or water deficit (Sect. 2.8.2). In the subsequent analysis, we do not consider hydraulic heads because the results almost entirely depend on the position of the observation well.

The stability of the relationship between drought indices and hydrological variables strongly depends on the chosen drought index and the irrigation scenario. In Fig. 6, we exemplify the relationship between SPEI and discharge for two irrigation scenarios in present and future climate. On the right panel of Fig. 6 (scenario FUTIRR), the relationship between SPEI and discharge is relatively stable in different climates. A drought with a similar intensity (as defined by SPEI) has similar impacts on discharge in present and future climate. On the left panel, the bias is larger. In this case, a drought with a particular SPEI-value

results in a different annual mean discharge in present and future climate.

As outlined above, we use two different performance metrics to quantify this bias, the model bias $B$ and the $NRMSE$ (Sect 2.8.2). Fig. 7 shows these two metrics for all indices and the two hydrological variables (discharge and water deficit) as bar plots. Overall, our results suggest that the relationships between the chosen meteorological drought indices and hydrological variables are not stable under a changing climate. The computed model biases between drought indices in present and future

climate appear important. In the scenario without irrigation, the largest observed model bias is 0.012 m$^3$/s for discharge and 3.1% for the water deficit (mean discharge in present climate: 0.015 m$^3$/s, mean annual water deficit: 80%). With irrigation, the largest bias for discharge is 0.006 m$^3$/s for the RAI drought index and 9% for water deficit (mean discharge: 0.03 m$^3$/s, mean annual water deficit for irrigated and non-irrigated zones: 52%). In the worst case described above (discharge without irrigation), the model bias can reach 80% of the value of the hydrological variable, which is a significant difference. For certain

conditions, however, the bias is low. For example, water deficit in the scenario without irrigation is predicted well by the linear model (the largest bias is equivalent to only 3.9% of the present water deficit).

For discharge, model bias depends strongly on the irrigation scenario (Fig. 7, top panels). With irrigation, the drought indices often underestimate the changes in discharge, especially if the indices are based on precipitation only. For example, in the case of SPI, the model bias for discharge is $-0.006$ m$^3$/s with irrigation (and 0.001 m$^3$/s without irrigation). On the contrary, drought

indices which are based on ET$_0$ and precipitation have a lower bias in the scenario with irrigation than in the scenario without irrigation. For example, SPEI has a model bias of 0.001 m$^3$/s with irrigation and of 0.012 m$^3$/s without irrigation. In the Lerma catchment, discharge is more sensitive to climate change when irrigation is present (von Gunten et al., 2015). Hence, drought indices which are more sensitive to climate change, notably to changes in ET$_0$, predict changes in discharge better in irrigated cases. The discharge in the scenario without irrigation does not change significantly and drought indices with a smaller reaction

to climate change are better predictors for hydrological impacts than those with a stronger reaction (Fig. 7, top panels).

For the water deficit (Fig. 7, bottom panels), drought indices which include ET$_0$ have a lower model bias than indices which only include precipitation. In the case of SPI with irrigation, the model bias is 8.5%. In the case of RDI, which includes ET$_0$, the model bias is 3.3%. The lower bias for drought indices containing ET$_0$ can be explained because ET$_0$ is directly influencing the water-deficit calculation.





The drought indices with the lowest model bias and a correlation coefficient $r > 0.6$ are: RAI for discharge in NOIRR scenario, RDI for the water deficit in FUTIRR/PIRR scenario, and SPEI for the water deficit in the NOIRR scenario and discharge in FUTIRR/PIRR scenario.

## 3.4 Future droughts

In Sect. 3.2 and Sect. 3.3, we explored the relationships between the different drought indices and the selected hydrological variables in present and future climate. In the present section, we compare the drought indices in present climate to those in future climate. This is a step forward compared to previous studies because we use the information of Sect. 3.2 and Sect. 3.3 to improve the predictions of future droughts, notably to interpret differences between the predictions based on different drought indices. Fig. 8 shows the changes between present and future climate in the seven drought indices based on the outputs of the four regional climate models. Note that a decrease in the values of the drought indices indicates an increase in drought intensity.

When we compare the changes in drought indices between present and future climate, significant differences can be observed between the different climate scenarios (based on the 4 regional climate models). Indices which only contain precipitation (RAI, SPI, and EDI) predict a small increase in droughts or a small decrease depending on the climate scenario (Fig. 8, top panels). For example, the average SPI decreases by -0.4 when using the ETHZ climate scenario and increases by 0.2 when using the MPI scenario (for comparison, an SPI of -3 would be an extreme drought). The MPI and UCLM regional climate models predict an increase in annual precipitation for the Lerma catchment (von Gunten et al., 2015). Hence, the climate scenarios based on these regional climate models result in a decrease in drought events (i.e., an increase in the drought index value) when indices are only based on precipitation. Indices which also consider $ET_0$ (Fig. 8, bottom panels) indicate an increase in droughts in all analyzed future climates. However, this increase is smaller when MPI and UCLM are used to construct the climate scenario. In the MPI case, a decrease of 1.1 in the mean value of SPEI is computed. When the ETHZ climate model is used, a decrease of 2.95 is computed (Fig. 8, bottom panel).

In addition to the differences related to the chosen climate scenario, the choice of the drought index has a large influence on the prediction of future droughts. These differences in drought prediction are largely the reflection of the differences in the linear relationships between drought indices and hydrological variables discussed in Sect. 3.3. If a drought index has a negative bias for discharge (as it is the case for indices which are based on precipitation only), small changes in future droughts are predicted. For example, when we average the four different climate scenarios, mean RAI in future climate shows a decrease of 0.02 when compared to RAI in present climate (Fig. 8, top panel, left column). Based on the linear model under present irrigation conditions, this can be translated to an increase in water deficit of 0.21 mm/year and a decrease in discharge of $8.7 \times 10^{-5}$ m$^3$/s. These changes are unlikely to have consequential impacts on irrigation or on the hydraulic regime of the catchment. For the indices that depend on $ET_0$, the predicted increase in droughts becomes larger. For example, mean SPEI shows a decrease of -2.43 (average of four regional climate models). If we would use the linear model developed in present climate, the decrease in discharge in the scenario with irrigation would be of 0.01 m$^3$/s, which is one third of the annual mean discharge. Based on the hydrological model, the change in discharge in the FUTIRR scenario is 0.006 m$^3$/s (average of the four climate models). Large uncertainties linked with climate prediction and hydrological modeling still prevail in this




estimation. However, the hydrological model generally reproduces discharge and hydraulic head measurements. Moreover, it simulates many relevant processes leading to discharge generation. Hence, we assess this model to be more reliable in predicting hydrological effects of climate change than a mere comparison of meteorological drought-index values.

## 4    Discussion

Outputs from global or regional climate models are often used to predict changes of droughts in future climates because these outputs are easy to obtain and relatively simple to analyze. In most cases, the analysis is based on the computation of meteorological drought indices. To use drought indices in climate-impact studies, it is necessary to choose a particular set of indices. Based on the assessment of correlation coefficients and stability of the relationships between hydrological variables and drought indices, the drought indices RDI, RAI, and SPEI are the most suitable indices in our case study. However, their

performance strongly depends on the assumed irrigation scenarios and may thus be different in other climates and land-uses. Other drought indices might perform better in more humid or colder climates. However, based on this study, these three indices are the most suitable for climate-impact studies in Mediterranean climate.

On a broader level, we propose to use drought indices with a certain caution in climate-impact studies and advise against using a single drought index. A hydrological model is a more direct way to analyze hydrological drought impacts in future

climate and it should be used whenever possible in such studies. Unfortunately, the development and the parameter calibration of hydrological models is a complicated task and depends on the availability of hydrological measurements such as discharge and hydraulic heads.

If the development of a hydrological model is not an option, our results suggest that outputs from drought indices should be analyzed in detail with respect to three issues, regardless of the set of the chosen drought indices:

1. The importance of potential evapotranspiration: Many meteorological drought indices only consider precipitation. Because these indices neglect the predicted increase in potential evapotranspiration, their uses could lead to an underestimation of future drought risks. This has been reported in previous studies, notably by Dubrovsky et al. (2009) and Zarch et al. (2015). Our study confirms that drought indices which neglect potential evapotranspiration predict smaller changes in droughts than those which include $ET_0$ (Sect. 3.4). However, we found that some indices that include $ET_0$, such as

SPEI, predict larger changes in drought severity compared to the simulations with the hydrological model (Sect 3.3), especially in scenarios with low soil moisture (scenario NOIRR). This was not previously considered and it indicates that, under some circumstances, the influence of $ET_0$ can be overestimated.

2. Correlation coefficients are not always sufficient to compare drought indices: Our comparison of the correlation coefficients between hydrological variables and drought indices (Sect 3.2) leads to similar results than previous studies. For

example, Vicente-Serrano et al. (2012a) compared the correlation between standardized stream flow (SSI) at monthly time scale and 6 drought indices, including SPI, SPEI, PDSI, and PHDI. SPEI showed the best correlation with discharge - results that we could reproduce (Fig. 5). SPI has a lower correlation than SPEI, but the difference is relatively small





in both studies. However, more detailed investigations of the relationships between the drought indices and hydrological variables provide new insights which are not possible to obtain by using correlation coefficients alone. For instance, the correlation coefficients between drought indices and annual mean discharge are similar in all scenarios and all climates within our study, while the regression coefficients change in future climate, and they do so differently in different irrigation scenarios. Hence, impacts of irrigation and climate on drought indices are better understood if we use analysis tools beyond correlation coefficients.

3. The hydrological impacts of drought depend on climate change: This has been previously explored in other studies, notably in studies focusing on hydrological droughts. For instance, Wanders et al. (2015) proposed a method to adapt the low-flow threshold defining the start of a hydrological drought as a function of the advance of climate change. The goal was to account for changes in the responses of low flows to droughts in a changing climate. However, these changes are also important when studying meteorological droughts. In this field, it is often assumed that the same lack of precipitation would have the same (hydrological) effects in present and future climate. However, this is not always the case (Sect 3.3). Investigating changes in frequency and intensity of meteorological droughts results in biased predictions of climate change impacts if changes in the hydrological processes are not considered.

## 5 Conclusions

The interpretation of changes in meteorological drought indices between future and present climates can be considerably compromised by the assumption that the relationship between the drought indices and the hydrological variables (which represent the effects of drought) is identical in present and future climate. The same drought-index value might lead to different drought consequences in present and future climate. Results can be further compromised by neglecting the increase in $ET_0$. In our case study, drought indices that take into account precipitation only (SPI, RAI, and EDI) underestimate the impact of droughts on water deficit and discharge often. By contrast, indices which give a high weight to $ET_0$ (as SPEI) sometimes overestimate the impact of future droughts on discharge, especially in the absence of irrigation.

As a summary, in the Lerma catchment, drought indices are useful indicators of dry periods in all tested climates and land-uses. However, a change in a particular drought index in future climate cannot easily be transferred to hydrological effects of droughts. In a stationary climate, the relationships between drought impacts and drought indices are usually reliable and so the hydrological consequences of droughts can be assessed from the drought indices. However, these relationships may change in a non-stationary climate and their evolution strongly depends on the particular combination of drought index and land-use. Hence, projections of future droughts using only one drought index may results in misleading estimation of the possible drought impacts.

Because drought indices can be estimated directly from the outputs of climate models, they are popular metrics of droughts even though they cannot be related uniquely to hydrological or even ecological impacts of droughts. Rather than relying on these indices, we recommend using a hydrological model to study hydrological effects of future droughts whenever possible. If setting up a hydrological model is not feasible, we advise to consider more than a single drought index and choose drought



indices that take both precipitation and $ET_0$ into account. We also advise to test the chosen drought indices against measured or modeled results.

Regardless of the chosen drought index or of the climate scenarios, this study, and many previous studies (e.g., Blenkinsop and Fowler, 2007), predict an increase in the severity of droughts in the next fifty years in northern Spain. Adaptation to the
new climatic conditions will therefore be necessary. The complexity of hydrological predictions should not prevent a timely adjustment of the urban water and irrigation networks.

## 6   Data availability

Hydrological data from the Lerma catchment have been collected and is owned by the Spanish Geological Survey (e.g., Merchán et al., 2013). Meteorological data have been collected by the Spanish meteorological national agency (AEMET) and is
currently proprietary. Data from the ENSEMBLES project is available at: http://ensemblesrt3.dmi.dk/.

*Acknowledgements.* We show our appreciation to H. Fowler and S. Blenkinsop for providing the weather generators and for their support. Moreover, we thank the Spanish meteorological national agency (AEMET) to provide us the meteorological data. In addition, we acknowledge the ENSEMBLES project, funded by the European Commission's 6th Framework Programme (contract number: GOCE-CT-2003-505539), to provide us the outputs from the regional climate models. Research in the Lerma catchment is supported by the European
Union (FEDER funds, grant CGL-2012-32395) of the Spanish Ministry of Economy and Competitiveness. The publication of this article is supported by the Deutsche Forschungsgemeinschaft and the" open access publishing fund" of the University of Tübingen. This study was performed within the International Research Training Group "Integrated Hydrosystem Modeling" (grant GRK 1829/1 of the Deutsche Forschungsgemeinschaft).





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





**Table 1.** A summary of the drought indices used in this study.

| Indices | Acronym | Input | Chosen time scale | Reference |
|---|---|---|---|---|
| Standardized precipitation index | SPI | P | 12 months | Svoboda et al. (2012) |
| Standardized precip. evapo. index | SPEI | P, PET | 12 months | Vicente-Serrano et al. (2009) |
| Rainfall anomaly index | RAI | P | 12 months | Keyantash and Dracup (2002) |
| Effective drought index | EDI | P | 12 months | Byun and Wilhite (1999) |
| Palmer drought severity index | PDSI | P, PET | $\sim$ 9 months | Palmer (1965) |
| Palmer hydrological drought index | PHDI | P, PET | $\sim$ 9 months | Palmer (1965) |
| Reconnaissance drought index | RDI | P, PET | 12 months | Tsakiris and Vangelis (2005) |

**Table 2.** Name and acronym of the regional climate models used in this study. - Adapted from Herrera et al. (2010) and von Gunten et al. (2015).

| Acronym | RCM | GCM | Reference |
|---|---|---|---|
| ETHZ | CLM | HadCM3 | Jaeger et al. (2008) |
| METO | HadRM3 | HadCM3 | Collins et al. (2006) |
| MPI | M- REMO | ECHAM5 | Jacob et al. (2001) |
| UCLM | PROMES | HadCM3 | Sánchez et al. (2004) |




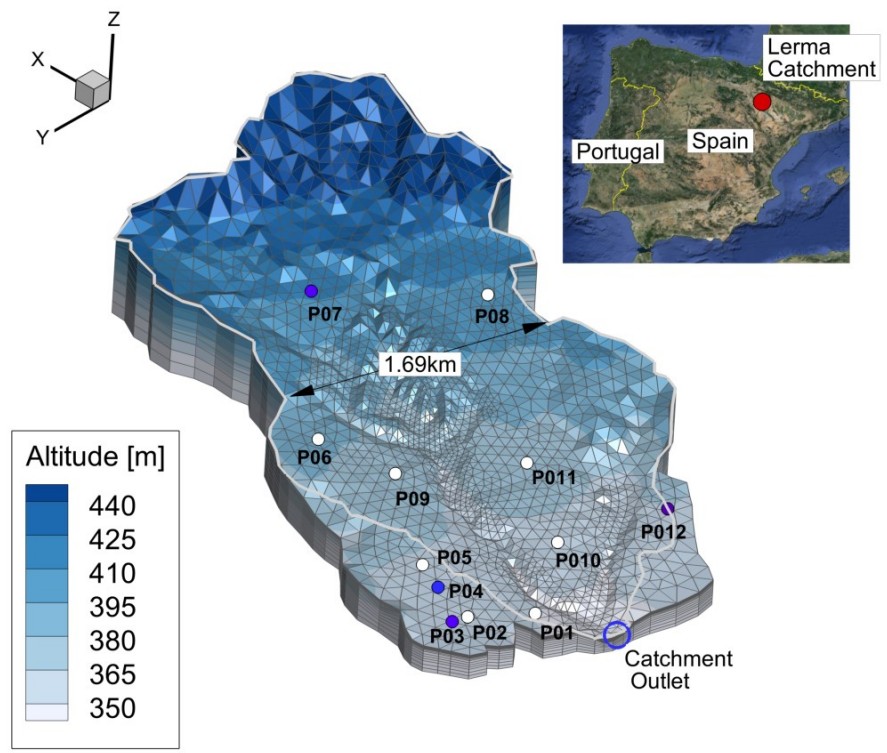

**Figure 1.** Surface elevation of the Lerma catchment (masl.). The observation wells drilled in 2010 are indicated by blue circles and the ones drilled in 2008 are indicated by white circles. The gray line represents the limits of the surface flow domain. Vertical exaggeration: 5:1. Modified from von Gunten et al. (2014, 2015).

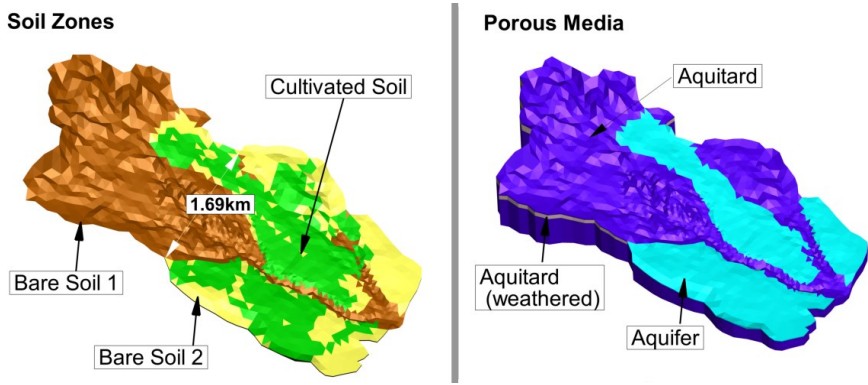

**Figure 2.** Soil and hydrogeological zones for the year 2009. Vertical exaggeration: 5:1. Modified from von Gunten et al. (2014, 2015).

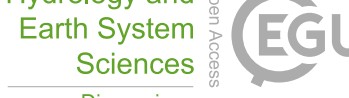



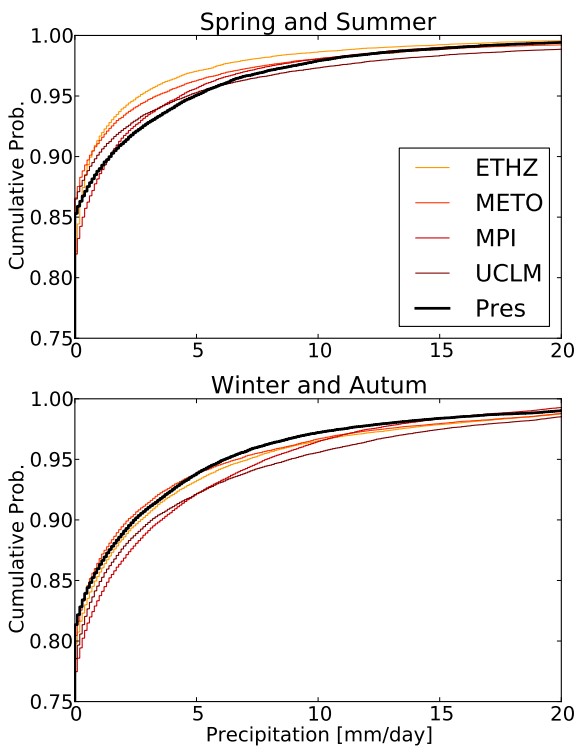

**Figure 3.** Empirical cumulative distribution function of daily precipitation for present and future climate scenarios.





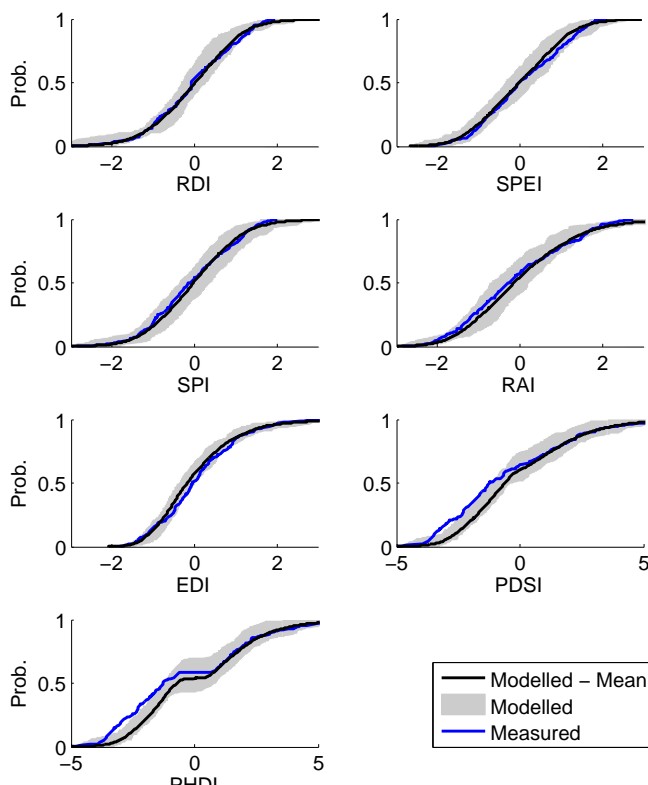

**Figure 4.** Empirical cumulative distribution function (*ecdf*) of drought indices based on measurement time series (in blue) and based on the outputs from the weather generator (in black). The gray area represents the boundaries of the 15 *ecdf* of drought indices based on the outputs from the weather generator when these outputs are cut at the same length that the measurement time series (23 years).




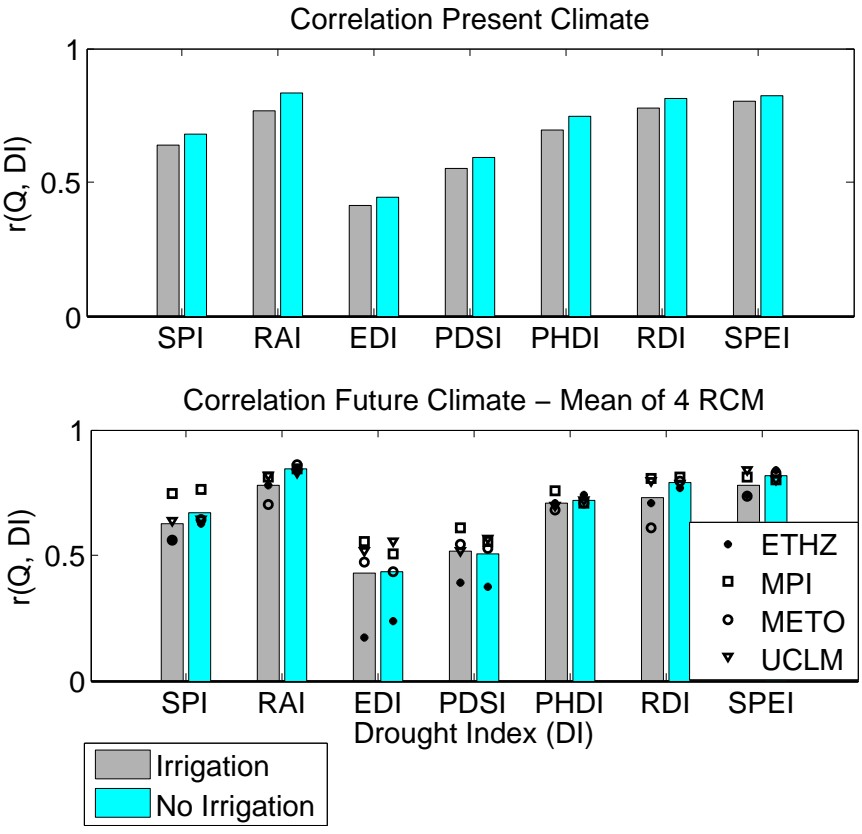

**Figure 5.** Correlation coefficient $r$ between the drought indices and discharge. In future climate (bottom panel), the plotted bars are the average of the outputs of the four regional climate models. See Table 2 for information about the four regional climate models.

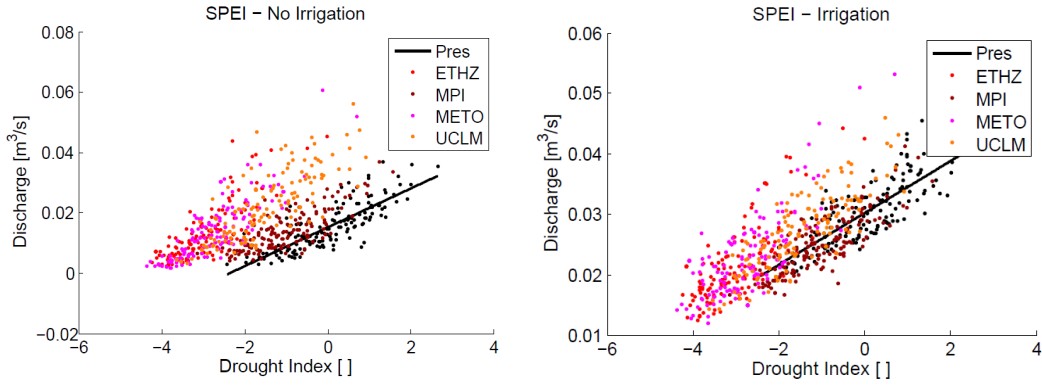

**Figure 6.** Performance of SPEI in future climate for annual discharge. The black line is the linear regression between SPEI and discharge in present climate. Left panel: NOIRR scenario, large model bias. Right panel: FUTIRR scenario, no significant model bias.





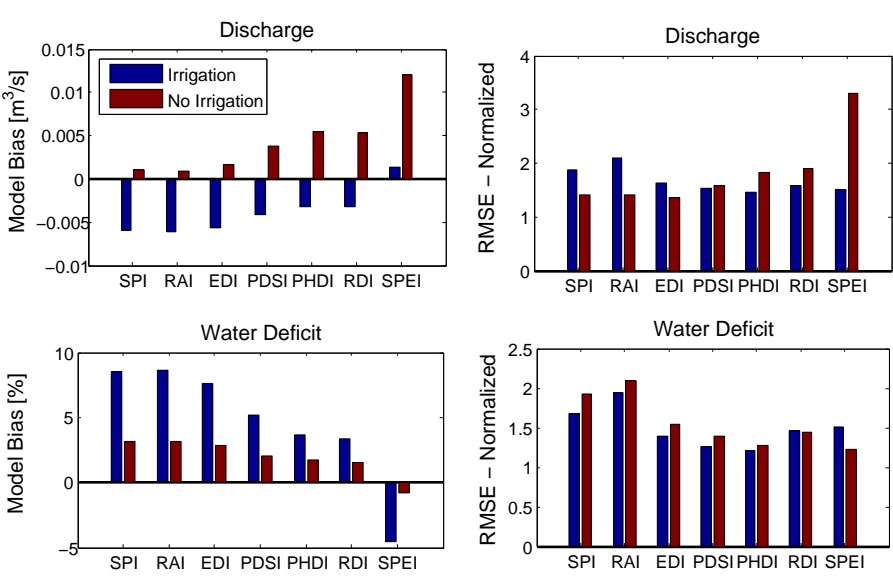

**Figure 7.** Model bias and NRMSE in the NOIRR and PIRR/FUTIRR irrigation scenarios.




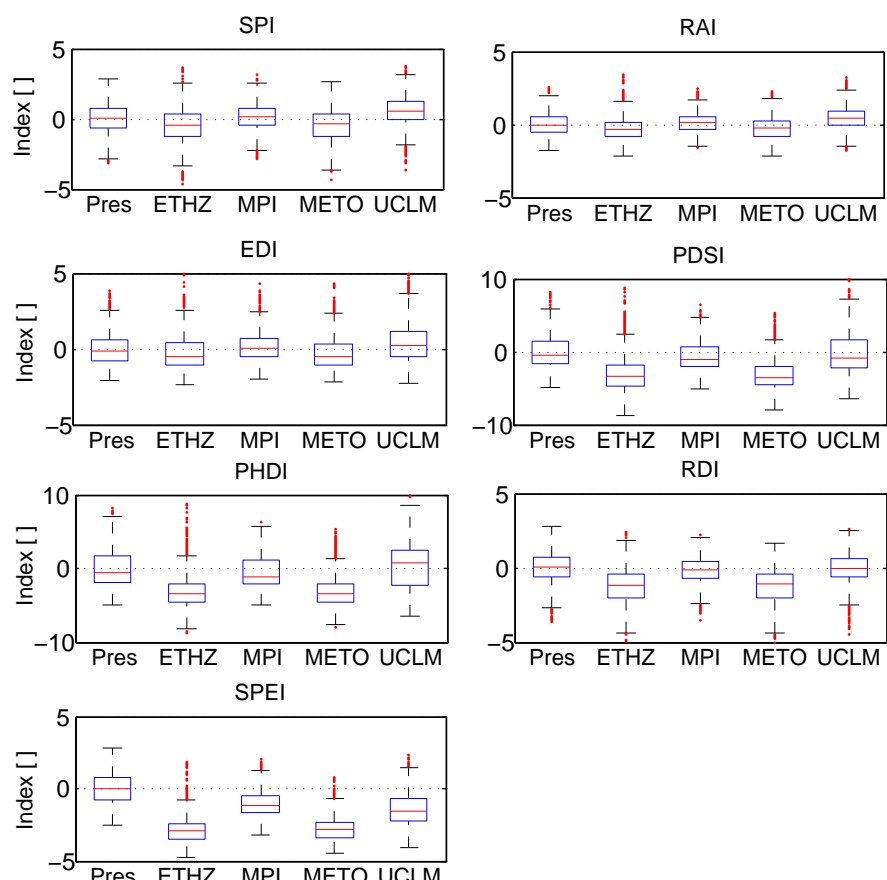

**Figure 8.** Present and future (2040-2050) droughts predicted by the seven drought indices, using the outputs from the weather generator. See Table 2 for information about the four regional climate models.