# Peer review of "Using an integrated hydrological model to estimate the usefulness of meteorological drought indices in a changing climate"

_Hydrology and Earth System Sciences, 2015_

## Referee Comment (RC1) · S. Bachmair (Referee) · 25 Feb 2016

General comments:

The study evaluates the suitability of seven meteorological drought indicators for assessing hydrological drought in present and future climates for a semiarid catchment in Spain. To assess the link between meteorological and hydrological drought in the future the authors use different climate scenarios for calculating drought indicators and compare it to discharge, water deficit, and groundwater heads simulated with a hydrological model. In addition, the authors investigate differences in future drought intensity based on different drought indicators and different regional climate models. The study addresses an important topic and reveals new insights into drought propagation with

regard to future change. The authors present interesting findings and draw valuable conclusions about the suitability of drought indicators.

One specific aspect the authors highlight is that correlation analysis between meteorological drought indicators and hydrological variables (as conducted in many studies) may not always provide sufficient information since it captures the dynamics only. Further measures should be considered (e.g. models bias). What I find somewhat contradictory in this respect is the low temporal resolution in the applied correlation and regression analysis (and therefore averaged dynamics). The authors use annual hydrological variables. I think that for drought analysis and better understanding drought propagation in the future a sub-annual resolution (at least seasonal dis-aggregation) would be highly desirable. Put differently, how much can we infer from an annual average relation between meteorological drought and streamflow or groundwater levels in contrast to e.g. seasonal data, especially when thinking about water management and planning issues? A concern that is somewhat related is that very little information is provided about hydrological processes in the catchment (now and changes in the future) and how they relate to differences in the linkage to drought indicators. Is enhanced ET the only factor? I would appreciate to see time series of modeled precipitation, ET, future discharge, groundwater levels etc. for a more process-based picture of the link between a precipitation decrease/ET increase and hydrological drought. This may also help to understand how generalizable the findings from this unique catchment are .

Regarding the paper presentation, the paper is well written and clearly structured. However, the manuscript would benefit from shortening the methods section (suggestions see below). Although I appreciate the attempt to be very transparent, currently almost 9 pages present methods, and only 5 results and discussion, which seems a bit imbalanced.

Detailed comments and/or suggestions:

Abstract:

"We conclude that meteorological drought indices are able to identify the timing of hydrological impacts of droughts in present and future climate. " I am bit concerned about the general inference on timing between the two variables looking at annual averages. What about e.g. "summer flash droughts" and intermittent heavy rainfall (likely leading to enhanced surface runoff and less recharge) versus a continuous seasonal dry period versus wetter period? Wouldn't the annual average response be similar, but the dynamics between meteorological and hydrological drought and thus water availability and implications for management be different at shorter time scales?

Methods:

Since you provide an overview of the methods in section 2.1 some of the later information is a bit redundant and could be heavily shortened.

P 5, L 6-13: Is this needed in this detail?

Study area: since you provide a detailed description of the basin a link to changes in catchment processes in the future may be interesting to pick up in the results/discussion.

Climate scenarios: Could this be shortened and potentially merged with the results 3.1 section (since this section contains quite a bit of methodology in my view)?

Irrigation scenarios: Where does irrigation water come from? Surface water, groundwater abstractions, reservoirs, are there any water transfers?

Drought indictors: This section could be strongly condensed. Do you really need the introductory part (P8, L13-30)? P, L23-30: this could go into the discussion section. SPI/SPEI/PDSI are all frequently used. I therefore suggest making reference to existing papers and keeping these methods brief.

Computation of potential evapotranspiration: Could some of the details go into an appendix?

Methods of comparing the drought indices to predict hydrological variables: Which model are the future drought indicators based on for predicting the hydrological response (e.g. shown in Figure 7)? I assume it is average of the outputs of the four regional climate models as in Figure 6 bottom panel but this information should be given in this section. I would suggest presenting a relative bias rather than an absolute one. In the results you also set the absolute values into context (e.g. P9, L21: "the largest bias is equivalent to only 3.9% of the present water deficit").

Results:

Section 3.1: see comment in methods section

Figure 3: There are seasonal differences, which is why I think information may be lost when only looking at annual averages for the correlation/regression analysis.

Section 3.2: P14, L13: "details are available in the supplementary material": where do I find this?

Figure 5: Is the irrigation scenario a mean of PIRR and FUTIRR or just PIRR? I don't fully agree that the correlation coefficients are all similar, as you write. How do you explain EDI < 0.5? I think if you decrease the panel size there would be enough space for including the correlation coefficients with water deficit and groundwater head. EDI performs especially poor when considering the ETHZ model – any ideas why?

Section 3.3: Generally I think this a very interesting approach and strong outcome. I consider figure 5-7 the core part of your results since this is where you address the link between meteorological indicators and hydrological variables in present and future climates. However, if you start out with three hydrological variables (including hydraulic heads) I would like to see this reflected in this section but currently there is no information about hydraulic heads in the presented material.

Figure 6, right panel: you write that "the relationship between SPEI and discharge is

relatively stable in different climates". I find it hard to distinguish the pink from the red dots but to me the slope of the pink or red dot relation looks higher than for the present regression line? Have you considered comparing/plotting regression coefficients for the different indicators and scenarios to go beyond this one SPEI example scatter plot?

Figure 7: see comment in methods on future drought indicators used for prediction: is this a multi-model average? Since you have different units for your hydrological variables and to better relate it to the present scenario I would prefer relative over absolute values for model bias. What about displaying model bias for groundwater head in Figure 7 in addition? What can you infer from the analysis of this variable?

Section 3.4: I am curious about the underlying drivers of the differences between models regarding drought intensity. It seems worthwhile to add some explanations into the discussion section.

General: To condense the results section you could omit a few sentences repeating/explaining methods or introducing figures since the figures are well readable (examples are: P14, L28-31; P15, L11-13).

---

## Referee Comment (RC2) · Anonymous Referee #2 · 31 Mar 2016

General Comments:

This study compares and contrast the reliability of seven meteorological drought indices to assess/predict the hydrological impacts of draughts for current and future climate in a semi-arid catchment in Spain. Not only climate scenarios projected by GCMs but also different irrigation scenarios are considered to evaluate these impacts. They compare drought indices produced based on GCMs with simulated hydrologic variables including discharge, groundwater levels and water deficit. At the bottom line, this study shows why do we need a hydrological model to study the hydrological impacts of future droughts. In case of unavailability of hydrological model, several drought indices should be used to our analyses, especially which take both precipitation and potential

evapotranspiration (PET). Again, future climate predictions not only result in changes in precipitation but also in PET. Hence inclusion of both variables in the drought index inhibits overestimation/underestimation of draught impacts.

I believe that the paper will be a good contribution to the Hydrology and Earth System Sciences. My major concern about the length of the Methods Section.

The paper is well written, clear, easy to follow and well structured. I enjoyed reading when all details are given clearly in the Methods Section. However, giving details ended up with a long Methods Sections. As seen, the Methods Section (Section 2) consists of 9 pages of the 19-page paper. Hence, one of the recommendation is moving the whole sub-parts of "Drought Indices (2.6.X)" the Appendix. Hence, if the reader wants to freshen his/her memory about these indices, then consult this Appendix. This moving will cause a two-page reduction in Methods Section.

I am not a strong proponent of any other recommendation on this part. These are my recommendations, the authors may (or not) follow these: - The formula of the Penman-Monteith equation may be given in the Appendix. - I am not sure how much do we need the details of the Person's correlation coefficient. If the authors want to give it, it may be given in the Appendix.

Finally, I do believe that some of these recommendations will shorten the Methods Section.

Minor Point:

I was curious about the current irrigation usage, and noticed that the irrigation usage is enormous. The irrigation from the Aragon River collected at the Yesa reservoir in 2011 is $2.1 \cdot 10^6$ m3. Size of the irrigated portion is 3.54 km2 from von Gunten et al. (2015). Hence the irrigation depth is 593 mm. On top of this number, mean annual precipitation (MAP) is $\sim$400 mm. The runoff, from Figure 6, with 0 SPEI, 2-3 m3/s which is equivalent to 23-35 mm for the entire basin. If I assume no deep drainage from

irrigation, water usage is roughly 1,000 mm per year. This number intrigued me in a lot. First, is this irrigation sustainable over the long-term period? ∼600 mm of irrigation within a 400 mm of MAP environment makes the farmers, ecosystem very dependent on this irrigation, or headwater sources, the Pyrenees. Secondly, this value seems somewhat upper limit for maximum irrigation. Because the ecosystem is approaching towards the PET which is 1300 mm. Another saying from water-limited to energy-limited.

I am not sure whether or not the authors agree with me, but I definitely encourage the authors write a few sentences into the Conclusion or the Discussion part about the sustainability of this current land-cover transformation. The demand for water due to PET changes of future climate (as seen drier outcomes of ETHZ) is much less significant than those of current land-cover transformation.

Minor Comments:

P4. L5. Wording. I recommend forcing only for meteorology. . . . .changes in meteorological forcing and land-use cover.

P8. L17. Please cite "Table 1" before citing "Table 2". It may be good to cite "Table 1" in Section 2.1. Or you may reorder Tables.

P19. L10. Please change . . . . project 'is' to . . . project 'are'. Data may use as a singular or plural, however in two previous sentences you used as plural, hence to ensure consistency. Figure 6. Can you ensure the y-scale similar for both figures? I think the limits are [0 0.08] or [0 0.07]. And definitely, y-value (discharge) must be truncate at zero. Morevoer, it needs a better colour selection.
* * *

---

## Author Comment (AC1) · 26 Apr 2016

We thank the reviewers for their relevant and useful comments. We are confident that we can adequately address each of these comments and that the revised paper will gain from the discussion. Please find below our response to each of the reviewers' comment.

**1   Comments of the reviewer 1**

1. **What I find somewhat contradictory in this respect is the low temporal res-**

olution in the applied correlation and regression analysis (and therefore av-
eraged dynamics). The authors use annual hydrological variables. I think
that for drought analysis and better understanding drought propagation
in the future a sub-annual resolution (at least seasonal dis-aggregation)
would be highly desirable. Put differently, how much can we infer from an
annual average relation between meteorological drought and streamflow
or groundwater levels in contrast to e.g. seasonal data, especially when
thinking about water management and planning issues?**

We are aware that seasonal and sub-annual scales are essential for drought pre-
diction and water management (e.g., Kumar et al., 2016). However, we have
conducted our analysis at the annual time scale in this paper because some of
the drought indices such PDSI or RDI cannot be directly applied at the seasonal
scale. Moreover, the central theme of this paper is the difference between the
correlations coefficients (which are similar in all studied climate scenarios) and
the model bias/RMSE (which depend on the climate and irrigation scenarios).
To study these differences, annual time scale is adequate in our point of view.
Indeed, a re-analysis at the seasonal time scale would not change our main con-
clusion (the need for a hydrological model) but it would unnecessarily lengthen
the paper. Finally, the annual timescale is used in many of the published papers
on future climate-change impacts (e.g., Kirono et al., 2011; Park et al., 2015), and
we wanted to ensure comparability with previous research efforts on this topic.
Hence, we have decided to present our analysis for the annual scale in this study.

2. **A concern that is somewhat related is that very little information is pro-
vided about hydrological processes in the catchment (now and changes in
the future) and how they relate to differences in the linkage to drought in-
dicators. Is enhanced ET the only factor? I would appreciate to see time
series of modeled precipitation, ET, future discharge, groundwater levels
etc. for a more process-based picture of the link between a precipitation**

**decrease/ET increase and hydrological drought. This may also help to understand how generalizable the findings from this unique catchment are.**

Thanks for this remark. Part of the information was provided in our recent paper (von Gunten et al., 2015). However, more information on local hydrological processes would help to clarify and interpret our results. We will provide more hydrological data in the revised version, such as precipitation, PET, Q, etc. In addition, we will add a discussion about how our findings are related to the hydrological regime and about the potential for generalization.

3. **Regarding the paper presentation, the paper is well written and clearly structured. However, the manuscript would benefit from shortening the methods section (suggestions see below). Although I appreciate the attempt to be very transparent, currently almost 9 pages present methods, and only 5 results and discussion, which seems a bit imbalanced.**

We agree with the reviewer. We will shorten the method section in the revision and move some paragraphs to the appendix.

4. **"We conclude that meteorological drought indices are able to identify the timing of hydrological impacts of droughts in present and future climate." I am bit concerned about the general inference on timing between the two variables looking at annual averages. What about e.g."summer flash droughts" and intermittent heavy rainfall (likely leading to enhanced surface runoff and less recharge) versus a continuous seasonal dry period versus wetter period? Wouldn't the annual average response be similar, but the dynamics between meteorological and hydrological drought and thus water availability and implications for management be different at shorter time scales?**

Our results are only valid for the annual timescale. We will reformulate the abstract (notably the highlighted sentence) to clarify this point. We have decided

to focus on drought effects at the annual time scales because it is the adequate timescale for analyzing long-term changes such as climate-change impacts (cf. answer to issue #1)

5. **Since you provide an overview of the methods in section 2.1 some of the later information is a bit redundant and could be heavily shortened.**

Yes, we will shorten the Section 2.1.

6. **P 5, L 6-13: Is this needed in this detail?**

We will shorten this part also.

7. **Study area: since you provide a detailed description of the basin a link to changes in catchment processes in the future may be interesting to pick up in the results/ discussion.**

A discussion on this topic was already presented in von Gunten et al. (2015). Hence, we kept this part brief to avoid repeating ourselves. We will nevertheless expand the discussion for relevant catchment processes.

8. **Climate scenarios: Could this be shortened and potentially merged with the results 3.1 section (since this section contains quite a bit of methodology in my view)?**

Yes, we will merge the two sections (3.1 and 2.4). We will also shorten Section 2.4 on the climate scenarios, and the method section in general.

9. **Irrigation scenarios: Where does irrigation water come from? Surface water, groundwater abstractions, reservoirs, are there any water transfers?**

The irrigation water is surface water from the Yesa reservoir, which is situated about 80 km north of the catchment, at the foot of the Pyrenees mountains. This information is already given in the introduction (p.4 line 3), but we can repeat it in

the paragraph on the irrigation scenarios. We did not assume any limitation on irrigation water.

10. **Drought indicators: This section could be strongly condensed. Do you really need the introductory part (P8, L13-30)? P, L23-30: this could go into the discussion section. SPI/SPEI/PDSI are all frequently used. I therefore suggest making reference to existing papers and keeping these methods brief.**

We agree with this comment. Consequently, we will move the description of the drought indices to the appendix. This will shorten the method section considerably.

11. **Computation of potential evapotranspiration: Could some of the details go into an appendix?**

Yes, we will shorten this part and move some of the information to the appendix.

12. **Methods of comparing the drought indices to predict hydrological variables: Which model are the future drought indicators based on for predicting the hydrological response (e.g. shown in Figure 7)? I assume it is average of the outputs of the four regional climate models as in Figure 6 bottom panel but this information should be given in this section.**

Yes, it is the average of the four climate models. We will add this information to the revision.

13. **I would suggest presenting a relative bias rather than an absolute one. In the results you also set the absolute values into context (e.g. P9, L21:"the largest bias is equivalent to only 3.9% of the present water deficit").**

We agree that presenting the relative bias makes our results more accessible. We will change the figure as proposed.

14. **Figure 3: There are seasonal differences, which is why I think information may be lost when only looking at annual averages for the correlation/regression analysis**

Yes, we agree that there are seasonal differences in the correlation coefficients. But our goal was not a general analysis of bias introduced by correlation coefficients. Please refer to our responses above regarding the choice of the annual time scale (answer to issue #1). We will nevertheless note this limitation in the revised text.

15. **Section 3.2: P14, L13:"details are available in the supplementary material": where do I find this?**

We sent the appendix with the paper. But we did not check if it was available. We will make sure that it is sent to you with the revision.

16. **Figure 5: Is the irrigation scenario a mean of PIRR and FUTIRR or just PIRR?**

We used the irrigation scenario PIRR in the present and FUTIRR in the future. We will clarify this in the label of the figure.

17. **I don't fully agree that the correlation coefficients are all similar, as you write.**

Thank you for pointing this out. This point was not described clearly enough. We meant that the correlation coefficients linked with a particular drought index were similar in the present and future climate, not that the correlation coefficients were similar for all drought indices. For example, the correlation coefficient between SPI and Q is similar for all climate/irrigation scenarios. But the correlation coefficient between EDI and Q is different than the one between SPI and Q. We will modify this paragraph to clarify our point.

18. **How do you explain EDI <0.5? EDI performs especially poor when considering the ETHZ model - any ideas why?**

We have various hypotheses, but they have not been investigated in detail. A possibility is that intense precipitation events, which are common in summer in the Lerma catchment, create outliners in the effective precipitation used by EDI. These outliners have large values, so a large impact on the correlation coefficients, but they have a low correlation with drought conditions, which could decrease the overall correlation. As requested, we will address this issue in the revised manuscript in more details after additional investigations.

19. **I think if you decrease the panel size there would be enough space for including the correlation coefficients with water deficit and groundwater head.**

We did try to plot all the correlation coefficients in the same figure before and it is indeed possible. We have provided this figure with all the correlation coefficients (Q, heads, and water deficit) in the appendix. So the proposed figure is part of the paper and the reader will have access to it. However, we did not include this figure in the main text because the figure is somewhat difficult to grasp in a short amount of time and because it would distract the reader from the major points of our study.

20. **If you start out with three hydrological variables (including hydraulic heads), I would like to see this reflected in this section but currently there is no information about hydraulic heads in the presented material.**

The correlation coefficients between hydraulic heads and drought indices strongly depend on the position of the wells (see the Figure 1 of the supplementary material). Based on our initial investigations, the model bias is also highly dependent on the well position. The hydraulic heads of one well can react very differently compared to the heads from another well. This makes the interpretation of the

various combinations of heads and drought indices complicated. Indeed, there are 12 wells and 7 drought indices. So we need to study 84 different cases to reach some conclusions and these conclusions would only be valid for these particular wells. Hence, the analysis might not be really useful for the reader. Consequently, we have decided to not analyze hydraulic heads further.

21. **Figure 6, right panel: you write that"the relationship between SPEI and discharge is relatively stable in different climates". I find it hard to distinguish the pink from the red dots but to me the slope of the pink or red dot relation looks higher than for the present regression line? Have you considered comparing/plotting regression coefficients for the different indicators and scenarios to go beyond this one SPEI example scatter plot?**

The goal of this figure is to show that the curves in the aforementioned case are stable *compared* to other cases. We wanted to illustrate the impact of the irrigation scenarios and the range of the possible outcome of our analysis. We will provide the regression coefficients in the revised version of the paper to simplify the comparison. In addition, the model bias (given in the next figure) can be used to compare the curve quantitatively. We will also modify the colors of the figure.

22. **Figure 7: Since you have different units for your hydrological variables and to better relate it to the present scenario I would prefer relative over absolute values for model bias.**

We will present the relative value of the model bias in the revised version. It will hopefully help the comparison.

23. **What about displaying model bias for groundwater head in Figure 7 in addition? What can you infer from the analysis of this variable?**

For the hydraulic heads, the main conclusion is that the response largely depends on the well localization (see issue #20). Therefore, we should show the

model bias for the 12 wells to provide accurate information. It would be a lot of information for one figure. Hence, we prefer to restrict our analysis to discharge and water deficit.

24. **Section 3.4: I am curious about the underlying drivers of the differences between models regarding drought intensity. It seems worthwhile to add some explanations into the discussion section.**

This is a very interesting question but a detailed analysis would go well beyond the scope of our study. Therefore, we will only add some comments on this subject in the discussion, based on the available literature.

25. **General: To condense the results section you could omit a few sentences repeating/ explaining methods or introducing figures since the figures are well readable (examples are: P14, L28-31; P15, L11-13).**

We will shorten these particular paragraphs.

**2  Comments of the reviewer 2**

1. **However, giving details ended up with a long Methods Sections. As seen, the Methods Section (Section 2) consists of 9 pages of the 19-page paper. Hence, one of the recommendation is moving the whole sub-parts of "Drought Indices (2.6.X)" in the Appendix.**

We agree with the reviewer. The method section is indeed too long. We will follow the suggestion to move the definition of the drought indices to the appendix and we will shorten our method section during the revision

2. **These are my recommendations, the authors may (or not) follow these:**

- **The formula of the Penman-Monteith equation may be given in the Appendix.**
- **I am not sure how much do we need the details of the Person's correlation coefficient. If the authors want to give it, it may be given in the Appendix.**

We will add the Penman-Monteith equation to the appendix. It is a central equation in this paper and it is therefore useful to ensure that the reader has access to it. We will also shorten the paragraph on the Pearson's correlation coefficient and move it to the appendix.

3. **I was curious about the current irrigation usage, and noticed that the irrigation usage is enormous. The irrigation from the Aragon River collected at the Yesa reservoir in 2011 is 2\* $10^6$ m$^3$. Size of the irrigated portion is 3.54 km$^2$ from von Gunten et al. (2015). Hence the irrigation depth is 593 mm. On top of this number, mean annual precipitation (MAP) is 400 mm. The runoff, from Figure 6, with SPEI, is 2-3 m$^3$/s which is equivalent to 23-35 mm for the entire basin. If I assume no deep drainage from irrigation, water usage is roughly 1,000 mm per year. This number intrigued me in a lot. First, is this irrigation sustainable over the long-term period? 600 mm of irrigation within a 400 mm of MAP environment makes the farmers, the ecosystem very dependent on this irrigation, or headwater sources, the Pyrenees. Secondly, this value seems somewhat upper limit for maximum irrigation. Because the ecosystem is approaching towards the PET which is 1300 mm. Another saying from water-limited to energy-limited. I am not sure whether or not the authors agree with me, but I definitely encourage the authors write a few sentences into the Conclusion or the Discussion part about the sustainability of this current land-cover transformation. The demand for water due to PET changes of future climate (as seen drier outcomes of ETHZ) is much less significant than those of current land-cover**

**transformation.**

Studying the impacts of the irrigation onset is a major topic of the current research in the Lerma catchment. It is obviously a very political, sensitive, and important issue, even if it is somewhat outside of the scope of this paper. We entirely agree that the impact of land-cover transformation has more impact locally than the impacts due to climate change (e.g., von Gunten et al., 2015). We also agree that deep drainage is usually small. Hence, the agriculture in the Lerma (and in the Bardenas region in general) depends on irrigation, and therefore on the headwaters from the Pyrenees. Moreover, the percentage of irrigated land is expected to further increase in the region and the Yesa reservoir is being modified to store more water. Hence, the regional agriculture will very strongly depend on the availability of irrigation water in the future. Is this sustainable? It largely depends on the future hydrologic conditions in the Pyrenees, particularly in the catchment of the Yesa reservoir, and on our estimation of the ecological need of the Aragon River (from which the irrigation water is diverted). However, in any case, irrigation in the future will need to be appropriately planned, and farmers will have to adapt, for example by changing the type of crops or by upgrading to more efficient irrigation systems. Hence, the sustainability of the system is questionable from our point of view. We will add a short note in the conclusion of this paper on this subject. But it would deserve a more in depth discussion, which would be outside of the subject of this particular paper.

4. **P4. L5. Wording. I recommend forcing only for meteorology.**

Thanks, we will modify this.

5. **P8. L17. Please cite"Table 1" before citing"Table 2". It may be good to cite"Table 1" in Section 2.1. Or you may reorder the Tables.**

Thanks, we will modify this.

6. **P19. L10. Please change ...project 'is' to ...project 'are'. Data may use as a singular or plural, however in two previous sentences you used as plural, hence to ensure consistency.**

   Thanks, we will modify this.

7. **Figure 6. Can you ensure the y-scale similar for both figures? I think the limits are [0 0.08] or [0 0.07]. And definitely, y-value (discharge) must be truncated at zero. Moreover, it needs a better colour selection.**

   Thanks, we will modify this.

**References**

Kirono, D., Kent, D., Hennessy, K., and Mpelasoka, F. *Characteristics of Australian droughts under enhanced greenhouse conditions: Results from 14 global climate models,* J. Arid Environ., 75, 566-575, 2011

Kumar, R., Musuuza, J. L., van Loon, A. F., Teuling, A. J., Barthel, R., Broek, J., Mai, J., Samaniego, L., and Attinger, S.. *Multiscale evaluation of the standardized precipitation index as a groundwater drought indicator*, Hydrol. Earth Syst. Sci. , 20, 1117-1131, 2016

Park, C.-K., Byun, H.-R., Deo, R., and Lee, B.-R. *Drought prediction till 2100 under RCP 8.5 climate change scenarios for Korea*, J. Hydrol., 526, 221-230, 2015

von Gunten, D., Wöhling, T., Haslauer, C., Merchán, D., Causapé, J., and Cirpka, O.A. *Estimating climate-change effects on a Mediterranean catchment under various irrigation conditions*, J. Hydrol. Reg. Stud., 4, 550-570, 2015.

---

## Author Response (AR1)

**Authors' response to the reviewers' comments**

-

Title: Using an integrated hydrological model to estimate the usefulness of meteorological drought indices in a changing climate

-

Submitted to: HESS. Manuscript number: hess-2015-510

Authors: D. von Gunten, T. Wöhling, C. Haslauer, D. Merchán, J. Causapé, O.A. Cirpka

July 3, 2016

We renew our thanks to the reviewers for their useful comments, which improved the quality and the clarity of the paper. We hope that the revised manuscript is now of adequate quality for publication in HESS. The reviewers' comments are addressed in detail below.

**1 Responses to the comments of reviewer 1**

The modifications related to the comments of the second reviewer are highlighted in blue in the manuscript.

1. **What I find somewhat contradictory in this respect is the low temporal resolution in the applied correlation and regression analysis (and therefore averaged dynamics). The authors use annual hydrological variables. I think that for drought analysis and better understanding drought propagation in the future a sub-annual resolution (at least seasonal dis-aggregation) would be highly desirable. Put differently, how much can we infer from an annual average relation between meteorological drought and streamflow or groundwater levels in contrast to e.g. seasonal data, especially when thinking about water management and planning issues?**

   We are aware that seasonal and sub-annual scales are essential for drought prediction and water management (e.g., Kumar et al., 2016). However, we have conducted our analysis at the annual time scale in this paper because some of the drought indices such PDSI or RDI cannot be directly applied at the seasonal scale. Moreover, the central theme of this paper is the difference between the correlations coefficients (which are similar in all studied climate scenarios) and the model bias/RMSE (which depend on the climate and irrigation scenarios). To study these differences, annual time scale is adequate in our point of view. Indeed, a re-analysis at

the seasonal time scale would not change our main conclusion (the need for a hydrological model) but it would unnecessarily lengthen the paper. Finally, the annual timescale is used in many of the published papers on future climate-change impacts (e.g., Kirono et al., 2011; Park et al., 2015), and we wanted to ensure comparability with previous research efforts on this topic. Hence, we have decided to present our analysis for the annual scale in this study. We have summarized our arguments in P5, line 10-15.

2. **A concern that is somewhat related is that very little information is provided about hydrological processes in the catchment (now and changes in the future) and how they relate to differences in the linkage to drought indicators. Is enhanced ET the only factor? I would appreciate to see time series of modeled precipitation, ET, future discharge, groundwater levels etc. for a more process-based picture of the link between a precipitation decrease/ET increase and hydrological drought. This may also help to understand how generalizable the findings from this unique catchment are.**

   We have added a subsection about the hydrological process in the Lerma catchment, focused on the impact of climate change on the hydrology (Sect. 3.5). The aim is to give to the reader a more general picture of the hydrology of the catchment in the different climate and irrigation scenarios. In addition, we have extended the section on future drought (Sect. 4.3) to analyze possible changes in the response of the catchment to droughts in present and future climate (P15, L19-35; P16, L1-4; and Figure 9). We have also included a short remark on the possible generalization of our result to other catchments in Sect. 5 (P16, L28-31).

3. **Regarding the paper presentation, the paper is well written and clearly structured. However, the manuscript would benefit from shortening the methods section (suggestions see below). Although I appreciate the attempt to be very transparent, currently almost 9 pages present methods, and only 5 results and discussion, which seems a bit imbalanced.**

   We have shortened the method section from 9 pages to 4.5 pages (not accounting for the subsection on the climate scenarios, which was displaced to Sect. 3).

4. **"We conclude that meteorological drought indices are able to identify the timing of hydrological impacts of droughts in present and future climate." I am bit concerned about the general inference on timing between the two variables looking at annual averages. What about e.g."summer flash droughts" and intermittent heavy rainfall (likely leading to enhanced surface runoff and less recharge) versus a continuous seasonal dry period versus wetter period? Wouldn't the annual average response be**

similar, but the dynamics between meteorological and hydrological drought and thus water availability and implications for management be different at shorter time scales?

We have noted in the abstract that we concentrated on the annual time scale (P1, L8) and we have reformulated the highlighted sentence to reflect this fact (P1, L18-19).

5. **Since you provide an overview of the methods in section 2.1 some of the later information is a bit redundant and could be heavily shortened.**

We have shortened Section 2.5 (Section 2.8 in the initial submission) from 40 lines to 31 lines.

6. **P 5, L 6-13: Is this needed in this detail?**

As we discuss the issue of hydrological versus meteorological droughts in the introduction (P2, L26-32; P3, L1-3), we have decided to remove this paragraph entirely.

7. **Study area: since you provide a detailed description of the basin a link to changes in catchment processes in the future may be interesting to pick up in the results/ discussion.**

We have added a figure (Fig. 9) and a paragraph to Section 4.3. In this paragraph, we develop our analysis of the differences in the catchment responses to droughts between present and future climate.

8. **Climate scenarios: Could this be shortened and potentially merged with the results 3.1 section (since this section contains quite a bit of methodology in my view)?**

We have merged the Sections 2.4, 2.5 and 3.1 into the Section 3 which now covers the irrigation and climate scenarios. We have slightly shortened the sections 2.4 and 3.1. The text in Section 2.4 (Section 3.1 in the revised version) is now six lines shorter than in the original version.

9. **Irrigation scenarios: Where does irrigation water come from? Surface water, groundwater abstractions, reservoirs, are there any water transfers?**

We have added a note on the provenance of the water in Section 3.3 which describes the irrigation scenarios (P11, L1-5). The irrigation water is imported from a reservoir outside of the catchment.

10. **Drought indicators: This section could be strongly condensed. Do you really need the introductory part (P8, L13-30)? P, L23-30: this could go into the discussion section. SPI/SPEI/PDSI are all frequently used. I therefore suggest making reference to existing papers and keeping these methods brief.**

We have moved the description of the drought indices to the appendix and shortened Section 2.6 from 82 lines in the original paper to 30 lines in the revised paper. We have also moved the paragraph on lines 23-30 to Section 4.3 (P14, L7-14).

11. **Computation of potential evapotranspiration: Could some of the details go into an appendix?**

We have moved this paragraph to the appendix. Only the principal information on potential evapotranspiration were kept in the main text (P6, L8-10 and P7, L26-28).

12. **Methods of comparing the drought indices to predict hydrological variables: Which**

**model are the future drought indicators based on for predicting the hydrological response (e.g. shown in Figure 7)? I assume it is average of the outputs of the four regional climate models as in Figure 6 bottom panel but this information should be given in this section.**

Yes, it is the average of the four climate models. This information was added to the legend of Figure 7.

13. **I would suggest presenting a relative bias rather than an absolute one. In the results you also set the absolute values into context (e.g. P9, L21:"the largest bias is equivalent to only 3.9% of the present water deficit").**

We agree that presenting the relative bias makes our results more accessible. We have changed the figure as proposed.

14. **Figure 3: There are seasonal differences, which is why I think information may be lost when only looking at annual averages for the correlation/regression analysis**

We have noted that our results are only valid at the annual scale on P12, L18-19.

15. **Section 3.2: P14, L13:"details are available in the supplementary material": where do I find this?**

The supplementary material should be available in the HESS website during the review. I will upload it again at least. Please send me a mail if it is still not online, I can also directly send the supplementary material to you.

16. **Figure 5: Is the irrigation scenario a mean of PIRR and FUTIRR or just PIRR?**

We have modified the label of Figure 5 to clarify the chosen irrigation scenario.

17. **I don't fully agree that the correlation coefficients are all similar, as you write.**

    This point was not described clearly enough. We meant that the correlation coefficients linked with a particular drought index were similar in the present and future climate, not that the correlation coefficients were similar for all drought indices. We have modified this paragraph to clarify our point on P12, L4-7.

18. **How do you explain EDI <0.5? EDI performs especially poor when considering the ETHZ model - any ideas why?**

    After further investigations, the relatively low correlation of EDI with yearly mean discharge and water deficit is probably related to the different weights accorded to the precipitation of each month.

    We can explain this in more detail by looking at the calculation of EDI. EDI is based on the normalization of effective precipitation $EP$ by:

    $$EP = \sum_{n=1}^{i} \frac{\sum_{d=1}^{n} P_d}{n} \tag{1}$$

    where $i$ is the summation period and $P_d$ is the precipitation of $d$ days before the end of the period $i$. We choose $i = 365$ days in our application. Based on equation (1), daily precipitation is not weighted equally. Precipitation which is close to the last day are given more weight than the precipitation of the day before. Indeed, $P_1$ is part of each term of the calculation of $EP$, while $P_i$ is only part of the last term. Practically, it means that precipitation in December is more important than precipitation in January for the calculation of EDI. However, precipitation in January is roughly of the same importance than precipitation in December for the hydrological variables. Hence, EDI is not well correlated with these variables.

    To illustrate this, we have modified the precipitation data. We artificially set the daily precipitation to zero in December, June, or February. We then looked at the influence of this change on the mean EDI. Average EDI is zero in the not-modified data as EDI is normalized on this dataset. EDI based on the precipitation data without rain in December shows the most important change.

    We have added this information to the supplementary material where the correlation coefficients are analyzed in detail (P6, L29-33).

[Figure]

Figure 1: mean EDI for original precipitation data, and precipitation data without rain in December, June, or February.

19. **I think if you decrease the panel size there would be enough space for including the correlation coefficients with water deficit and groundwater head.**

    We did try to plot all the correlation coefficients in the same figure before and it is indeed possible. We have provided this figure with all the correlation coefficients (Q, heads, and water deficit) in the appendix. So the proposed figure is part of the paper and the reader will have access to it. However, we did not include this figure in the main text because the figure is somewhat difficult to grasp in a short amount of time and because it would distract the reader from the major points of our study.

20. **If you start out with three hydrological variables (including hydraulic heads), I would like to see this reflected in this section but currently there is no information about hydraulic heads in the presented material.**

    The correlation coefficients between hydraulic heads and drought indices strongly depend on the position of the wells (see the Figure 1 of the supplementary material). Based on our initial investigations, the model bias is also highly dependent on the well position. The hydraulic heads of one well can react very differently compared to the heads from another well. This makes the interpretation of the various combinations of heads and drought indices complicated. Indeed, there are 12 wells and 7 drought indices. So we need to study 84 different cases to reach some conclusions and these conclusions would only be valid for these particular wells. Hence,

the analysis might not be really useful for the reader. Consequently, we have decided to not analyze hydraulic heads further.

21. **Figure 6, right panel: you write that"the relationship between SPEI and discharge is relatively stable in different climates". I find it hard to distinguish the pink from the red dots but to me the slope of the pink or red dot relation looks higher than for the present regression line? Have you considered comparing/plotting regression coefficients for the different indicators and scenarios to go beyond this one SPEI example scatter plot?**

    We have added regression coefficients for the SPEI case in the main paper to provide a more quantitative way of comparing the two figures. We have also modified the colors of this figure. In addition, we provide below two figures which show the regression coefficients for all the drought indices for discharge and water deficit.

22. **Figure 7: Since you have different units for your hydrological variables and to better relate it to the present scenario I would prefer relative over absolute values for model bias.**

    We now use the relative model bias instead of the absolute bias in our analysis.

23. **What about displaying model bias for groundwater head in Figure 7 in addition? What can you infer from the analysis of this variable?**

    For the hydraulic heads, the main conclusion is that the response largely depends on the well localization (see issue #20). Therefore, we should show the model bias for the 12 wells to provide accurate information. It would be a lot of information for one figure. Hence, we prefer to restrict our analysis to discharge and water deficit.

24. **Section 3.4: I am curious about the underlying drivers of the differences between models regarding drought intensity. It seems worthwhile to add some explanations into the discussion section.**

    We have added some comments on this question in Section 4.3 (P14, L32-34 and P15, L1-2).

25. **General: To condense the results section you could omit a few sentences repeating/ explaining methods or introducing figures since the figures are well readable (examples are: P14, L28-31; P15, L11-13).**

    We have shortened the second paragraph on P13, L5-6. (P15, L11-13 in the initial submission).

[Figure]

Figure 2: Coefficients of the linear regression of discharge and drought indices for the present climate and the four climate scenarios.

[Figure]

Figure 3: Coefficients of the linear regression of water deficit and drought indices for the present climate and the four climate scenarios.

**2 Responses to the comments of reviewer 2**

The modifications related to the comments of the second reviewer are highlighted in red in the manuscript.

1. **However, giving details ended up with a long Methods Sections. As seen, the Methods Section (Section 2) consists of 9 pages of the 19-page paper. Hence, one of the recommendation is moving the whole sub-parts of "Drought Indices (2.6.X)" in the Appendix.**

   We have moved this paragraph to the appendix and we now only give a short introduction to the selected drought indices in Section 2.6. The length of the Section 2.6 (now Section 2.4) was 82 lines in the original paper. It is now reduced to 30 lines.

2. **These are my recommendations, the authors may (or not) follow these:**

   - **The formula of the Penman-Monteith equation may be given in the Appendix.**
   - **I am not sure how much do we need the details of the Person's correlation coefficient. If the authors want to give it, it may be given in the Appendix.**

   We have added the Penman-Monteith equation to the appendix (Section 2 of the appendix). We have also shortened the paragraph on the Pearson's correlation coefficient (from 15 lines to 9 lines in the revised version). However, we did not move it to the appendix because of the importance of the Pearson coefficient in this paper.

3. **I was curious about the current irrigation usage, and noticed that the irrigation usage is enormous. The irrigation from the Aragon River collected at the Yesa reservoir in 2011 is 2* $10^6$ m$^3$. Size of the irrigated portion is 3.54 km$^2$ from von Gunten et al. (2015). Hence the irrigation depth is 593 mm. On top of this number, mean annual precipitation (MAP) is 400 mm. The runoff, from Figure 6, with SPEI, is 2-3 m$^3$/s which is equivalent to 23-35 mm for the entire basin. If I assume no deep drainage from irrigation, water usage is roughly 1,000 mm per year. This number intrigued me in a lot. First, is this irrigation sustainable over the long-term period? 600 mm of irrigation within a 400 mm of MAP environment makes the farmers, the ecosystem very dependent on this irrigation, or headwater sources, the Pyrenees. Secondly, this value seems somewhat upper limit for maximum irrigation. Because the ecosystem is approaching**

**towards the PET which is 1300 mm. Another saying from water-limited to energy-limited. I am not sure whether or not the authors agree with me, but I definitely encourage the authors write a few sentences into the Conclusion or the Discussion part about the sustainability of this current land-cover transformation. The demand for water due to PET changes of future climate (as seen drier outcomes of ETHZ) is much less significant than those of current land-cover transformation.**

Studying the impacts of the irrigation onset is a major topic of the current research in the Lerma catchment. It is obviously a very political, sensitive, and important issue, even if it is somewhat outside of the scope of this paper. We entirely agree that the impact of land-cover transformation has more impact locally than the impacts due to climate change (e.g., von Gunten et al., 2015). We also agree that deep drainage is usually small. Hence, the agriculture in the Lerma (and in the Bardenas region in general) depends on irrigation, and therefore on the headwaters from the Pyrenees. Moreover, the percentage of irrigated land is expected to further increase in the region and the Yesa reservoir is being modified to store more water. Hence, the regional agriculture will very strongly depend on the availability of irrigation water in the future. Is this sustainable? It largely depends on the future hydrologic conditions in the Pyrenees, particularly in the catchment of the Yesa reservoir, and on our estimation of the ecological need of the Aragon River (from which the irrigation water is diverted). However, in any case, irrigation in the future will need to be appropriately planned, and farmers will have to adapt, for example by changing the type of crops or by upgrading to more efficient irrigation systems. Hence, the sustainability of the system is questionable from our point of view. We have added a short comment on irrigation management at the end of the conclusion (P18, L12-13). But it would deserve a more in-depth discussion, which would be outside of the subject of this particular paper.

4. **P4. L5. Wording. I recommend forcing only for meteorology.**

   This has been modified on P4, L5.

5. **P8. L17. Please cite"Table 1" before citing"Table 2". It may be good to cite"Table 1" in Section 2.1. Or you may reorder the Tables.**

   We now refer to Table 1 in the introduction in P3, L25.

6. **P19. L10. Please change ...project is' to ...project are'. Data may use as a singular or plural, however in two previous sentences you used as plural, hence to ensure consistency.**

   This has been modified on P18, L17.

7. **Figure 6. Can you ensure the y-scale similar for both figures? I think the limits are [0 0.08] or [0 0.07]. And definitely, y-value (discharge) must be truncated at zero. Morevoer, it needs a better colour selection.**

Figure 6 has been modified as proposed.

**References**

[revised manuscript text omitted]

---

## Referee Report (RR1)

Many thanks to the authors for their great effort to address reviewers' comments. I recommend the paper for publication after minor editing process which mostly to enhance the clarity and readability of the manuscript.

Minor comment:

P11. L21. I did not understand the following part of the sentence:

'… because the dryer soil results in lower infiltration during thunderstorms'.

I think dryer soils have greater infiltration capacity than wet soils, remember Horton infiltration capacity. Infiltration capacity decays with saturation, and finally reach at 'saturated hydraulic value'.

Maybe, I am missing one point here! Possible explanations in my mind are:

- Dryer or bare soils due to less or no vegetation may have lower infiltration capacity than vegetated soils.
- Or forming a soil crust that inhibit infiltration?

The sentence part from the text does not sound logical to me!

Minor Points Regarding the Text:

P2. L24. This sentence sounds a plural sentence. It should be:

Significant CORRELATIONS between … and … WERE found.

P2. L25. Can you tell the reason with one word or a few words? Hence, you clarify the reason of the phenomenon.

The correlation between groundwater levels and drought indices seems to be smaller than for other drought impacts due to XXXX, but it was still noticeable (Kumar et al., 2016).

P2. L32. Wording. I think there is a missing word.

…. are often two perspectives…

My recommendation: ….are often ASSESSED two perspectives of the same drought event.

P4. L6. Wording. I recommend following instead of using 'test':

…schemes to COMPARE THE RESPONSE OF different drought indices.

P4. L26. I recommend you to include no water-limitation. My recommendation is:

…under standard conditions with no soil moisture limitation (Allen et al., 1998).

P4. L26. Please change the year of citation Allen et al., to Allen et al., 1998. FAO56 is from 1998.

You can check: http://www.fao.org/docrep/X0490E/X0490E00.htm

P4. L27. Wording. I may misunderstood here! My understanding is AET should be calculated daily timescale based on $ET_o$ (Eq9 in Supp.). My suggestion:

…calculated on a daily time scale, and aggregated for each year.

If I am wrong, please add one sentence to describe how to calculate AET.

P6. L4. It should be plural. You are finding more than one values. COEFFICIENTS.

P5. L21-22. Wording. Radiation should be "incoming solar radiation", or "solar irradiance". Wind SPEED. My recommendation:

Solar irradiance, wind speed, and relative humidity have been measured.

P5. L21-22. Wording. You already define "annual precipitation", hence mm/year should be "mm". Replace mm/year ➔ mm.

P5. L25. My recommendation. Convert volume into "mm/year". This conversion will help the reader to understand faster. As you did this in P15 L9.

P5. L26. Wording. My recommendation, word order. If you put crop types in advance, the flow of the sentence will be better due to the transition from annual to daily time scale, also closeness of hydrological variables. My suggestion:

… crop types, monthly hydraulic head data, daily discharge and irrigation volume are available.

P6. L17. Wording. This is a recommendation. Applying totally depends on authors. I prefer using "anisotropic" to define the phenomenon. My suggestions:

 … the saturated hydraulic conductivity is anisotropic which horizontal one is one order greater than vertical one.

P6. L21. Wording. The word "Data" is plural! Singular form of 'data' is 'datum'. Yes, many sources may use as singular. It is OK. This is on discretion of the authors!

P6. L21. I recommend using numbers less than ten as word. If you disagree, be consistent within the text. As example: 4 models (P14. L18) or four models (P15. L7).

… a three-hour mean….     …… a nine-hour mean…..

P7. L 17. Insert a space after periods.

P7. L23. Delete 'only'. The exceptions are….

P8. L14.  I think this should be index.

… the same drought INDEX in future climate.

P9. L4. I think this should be plural. Climate scenarios.

P10. L16. …. a TWO-sided Kolmogorov-Smirnov…

P10. L22. Insert a COMMA after the reference. The second sentence is an independent clause and requires a comma.

… (e.g., Burton et al., 2010), and ….

P11. L2. Recommendation. I recommend giving mm/year in parenthesis. It is up to the authors!

P11. L7. Recommendation. I highly recommend using a conjunction at the beginning of this sentence to emphasize the different behaviour.

On the contrary, in winter and autumn, …..

P12. L 21. This sentence is should be plural. Is not it? You showed many linear correlation values between drought indices and hydrological values.

P12. L 20. I recommend delete the negative sign before 0.4.

… decreases by 0.4 when….

P15. L12. Similar above. You already talk about a decrease. No need a negative sign before 2.43.

P15. L19. Replace an with a.

…… with A SPI- and A SPEI-value

P16 L 13, L30: Use a lowercase when defining mediterranean climate.

… the mediterranean climate…

I know that currently having a trouble when writing this text, MS Office automatically correct it and underlines it as an error. But when defining the climate use lower case.

P17. L2. Wording. Similar TO.

…leads to similar results TO previous studies.

P17. L 4. …and SIX drought indices.

P17. L 28. Wording preference. I recommend:

… in all tested climate SCENARIOS and land-uses.

P18. L 15. …have been collected and ARE owned…

P18. L16. …. and ARE currently proprietary.

Figure 7. Can you shift the legend box to left in the top right figure?

---

## Author Response (AR2)

**Authors' response to the reviewers' comments**

-

Title: Using an integrated hydrological model to estimate the usefulness of
meteorological drought indices in a changing climate

-

Submitted to: HESS. Manuscript number: hess-2015-510

Authors: D. von Gunten, T. Wöhling, C. Haslauer, D. Merchán, J. Causapé, O.A. Cirpka

August 17, 2016

We thank the reviewer for his/her useful and detailed comments. We hope that the revised manuscript is now of adequate quality for publication in HESS. The reviewer's comments are addressed in detail below.

**1 Responses to the comments of reviewer**

The modifications related to the comments of the reviewer are highlighted in red in the manuscript.

1. **P11. L21. I did not understand the following part of the sentence: because the dryer soil results in lower infiltration during thunderstorms. I think dryer soils have greater infiltration capacity than wet soils, remember Horton infiltration capacity. Infiltration capacity decays with saturation, and finally reach at saturated hydraulic value.Maybe, I am missing one point here! Possible explanations in my mind are: - Dryer or bare soils due to less or no vegetation may have lower infiltration capacity than vegetated soils. - Or forming a soil crust that inhibit infiltration? The sentence part from the text does not sound logical to me!**

   Yes, this point was not explained clearly enough. We have discussed it in more detail in Section 6.4 of our preceding paper (von Gunten et al., 2015). Summarily, the start of irrigation has resulted in lower discharges and higher infiltration rates during intense rain events. This was observed in modelled and measured time series of discharge. Reasons for this phenomena include the changes in soil compaction due to agriculture, the higher soil moisture in the deeper soil layers (which modifies the unsaturated hydraulic conductivity and therefore the infiltration from the saturated surface layers), and a more compact vegetation which has decreased the surface flow velocity, resulting in higher contact times and higher infiltration during rain events. In the revised version of our paper,

we have highlighted the influence of vegetation on the infiltration rate, instead of the soil moisture (P11, L21-22), to improve the clarity of the text.

2. **P2. L24. This sentence sounds a plural sentence. It should be: Significant CORRELATIONS between  and  WERE found.**

   This has been modified as suggested on P2, L24.

3. **P2. L25. Can you tell the reason with one word or a few words? Hence, you clarify the reason of the phenomenon. The correlation between groundwater levels and drought indices seems to be smaller than for other drought impacts due to XXXX, but it was still noticeable (Kumar et al., 2016).**

   One of the main reasons for the relatively low correlation between groundwater levels and drought indices is linked with the spatial and temporal variations of unsaturated hydraulic conductivity. Because of these variations, the time needed for the rain to reach the water table shows important variations (Kumar et al., 2016), even for the same aquifer. Hence, the impacts of droughts on groundwater are more complicated to analyze than the impacts on discharge, for example. We have added this information on P2, L25-26.

4. **P2. L32. Wording. I think there is a missing word. . are often two perspectives My recommendation: .are often ASSESSED two perspectives of the same drought event.**

   This has been modified as suggested on P2, L32

5. **P4. L6. Wording. I recommend following instead of using test: schemes to COMPARE THE RESPONSE OF different drought indices.**

   This has been modified as suggested on P4, L6.

6. **P4. L26. I recommend you to include no water-limitation. My recommendation is: under standard conditions with no soil moisture limitation (Allen et al., 1998).**

   This has been modified as suggested on P4, L26.

7. **P4. L26. Please change the year of citation Allen et al., to Allen et al., 1998. FAO56 is from 1998. You can check:**

   **http://www.fao.org/docrep/X0490E/X0490E00.htm**

   Thanks for noticing this. The citation was modified in the bibliography.

8. **P4. L27. Wording. I may misunderstood here! My understanding is AET should be calculated daily timescale based on ETo (Eq9 in Supp.). My suggestion: calculated on a daily time scale,**

**and aggregated for each year. If I am wrong, please add one sentence to describe how to calculate AET.**

Yes, AET is calculated on the daily time scale and is aggregated for each year. This has been modified as suggested on P4, L27.

9. **P6. L4. It should be plural. You are finding more than one values. COEFFICIENTS.**

This has been modified as suggested on P5, L4.

10. **P5. L21-22. Wording. Radiation should be incoming solar radiation, or solar irradiance. Wind SPEED. My recommendation: Solar irradiance, wind speed, and relative humidity have been measured.**

This has been modified as suggested on P5, L21-22.

11. **P5. L21-22. Wording. You already define annual precipitation, hence mm/year should be mm. Replace mm/year mm.**

This has been modified as suggested on P5, L22.

12. **P5. L25. My recommendation. Convert volume into mm/year. This conversion will help the reader to understand faster. As you did this in P15 L9.**

This has been modified as suggested on P5, L25.

13. **P5. L26. Wording. My recommendation, word order. If you put crop types in advance, the flow of the sentence will be better due to the transition from annual to daily time scale, also closeness of hydrological variables. My suggestion: crop types, monthly hydraulic head data, daily discharge and irrigation volume are available.**

This has been modified as suggested on P5, L26.

14. **P6. L17. Wording. This is a recommendation. Applying totally depends on authors. I prefer using anisotropic to define the phenomenon. My suggestions: the saturated hydraulic conductivity is anisotropic which horizontal one is one order greater than vertical one.**

We have modified the highlighted sentence on P6, L17 to add the word anisotropy to our description of the hydraulic conductivity.

15. **P6. L21. Wording. The word Data is plural! Singular form of data is datum. Yes, many sources may use as singular. It is OK. This is on discretion of the authors!**

This has been modified as suggested on P6, L21.

16. **P6. L21. I recommend using numbers less than ten as word. If you disagree, be consistent within the text. As example: 4 models (P14. L18) or four models (P15. L7). a three-hour mean. a nine-hour mean..**

    This has been modified as suggested on P6, L21 and on P14, L21.

17. **P7. L 17. Insert a space after periods.**

    This has been modified as suggested on P7, L17.

18. **P7. L23. Delete only. The exceptions are.**

    This has been modified as suggested on P7, L23.

19. **P8. L14. I think this should be index. the same drought INDEX in future climate.**

    This has been modified as suggested on P8, L14.

20. **P9. L4. I think this should be plural. Climate scenarios.**

    This has been modified as suggested on P9, L4.

21. **P10. L16. . a TWO-sided Kolmogorov-Smirnov**

    This has been modified as suggested on P10, L16.

22. **P10. L22. Insert a COMMA after the reference. The second sentence is an independent clause and requires a comma. (e.g., Burton et al., 2010), and .**

    This has been modified as suggested on P10, L22.

23. **P11. L2. Recommendation. I recommend giving mm/year in parenthesis. It is up to the authors!**

    This has been modified as suggested on P11, L3.

24. **P11. L7. Recommendation. I highly recommend using a conjunction at the beginning of this sentence to emphasize the different behaviour. On the contrary, in winter and autumn, ..**

    This has been modified as suggested on P11, L7.

25. **P12. L 21. This sentence is should be plural. Is not it? You showed many linear correlation values between drought indices and hydrological values.**

    This has been modified as suggested on P12, L25.

26. **P12. L 20. I recommend delete the negative sign before 0.4. decreases by 0.4 when.**

    This has been modified as suggested on P14, L23.

27. **P15. L12. Similar above. You already talk about a decrease. No need a negative sign before 2.43.**

    This has been modified as suggested on P15, L15.

28. **P15. L19. Replace an with a. with A SPI- and A SPEI-value**

    This has been modified as suggested on P15, L22.

29. **P16 L 13, L30: Use a lowercase when defining mediterranean climate. the mediterranean climate I know that currently having a trouble when writing this text, MS Office automatically correct it and underlines it as an error. But when defining the climate use lower case.**

    This has been modified as suggested on P16, L16 and P17, L1.

30. **P17. L2. Wording. Similar TO. leads to similar results TO previous studies.**

    This has been modified as suggested on P17, L4.

31. **P17. L 4. and SIX drought indices.**

    This has been modified as suggested on P17, L6.

32. **P17. L 28. Wording preference. I recommend: in all tested climate SCENARIOS and land-uses.**

    This has been modified as suggested on P17, L30.

33. **P18. L 15. have been collected and ARE owned**

    This has been modified as suggested on P18, L15.

34. **P18. L16. . and ARE currently proprietary.**

    This has been modified as suggested on P18, L15.

35. **Figure 7. Can you shift the legend box to left in the top right figure?**

    The legend of Figure 7 has been moved as suggested.

**References**

[revised manuscript text omitted]